# SINGLE-VIEW 3D-AWARE REPRESENTATIONS FOR REINFORCEMENT LEARNING BY CROSS-VIEW NEURAL RADIANCE FIELDS

## ABSTRACT

Reinforcement learning (RL) has enabled robots to develop complex skills, but its success in image-based tasks often depends on effective representation learning. Prior works have primarily focused on 2D representations, often overlooking the inherent 3D geometric structure of the world, or have attempted to learn 3D representations that require extensive resources such as synchronized multi-view images even during deployment. To address these issues, we propose a novel RL framework that extracts 3D-aware representations from single-view RGB input, without requiring camera pose or synchronized multi-view images during the downstream RL. Our method employs an autoencoder architecture, using a masked ViT as the encoder and a latent-conditioned NeRF as the decoder, trained with cross-view completion to capture fine-grained, 3D geometry-aware representations. Additionally, we utilize a time contrastive loss that further regularizes the learned representation for consistency across different viewpoints. Our method significantly enhances the RL agent's performance in complex tasks, demonstrating superior effectiveness compared to prior 3D-aware representation-based methods, even when using only single-view RGB images during deployment.

## 1 INTRODUCTION

Reinforcement learning (RL) has empowered an embodied agent such as a robot to acquire complex skills. However, its capability heavily depends on the representation of the underlying systems, especially in the image domain. In other words, central to image-based RL is the challenge of representation learning, where the goal is to distill high-dimensional visual data into compact, informative features that capture the essence of the environment. Effective representation learning schemes for image-based RL enable agents to interpret and act upon visual data more efficiently, facilitating faster convergence and improving performance in tasks such as robotic manipulation.

Many previous works have focused on learning efficient representation from visual inputs for downstream RL tasks. These can be roughly categorized into several approaches: pre-training an image encoder via contrastive objectives (Laskin et al., 2020; Nair et al., 2022), employing data augmentations (Yarats et al., 2021a;c), using autoencoders for reconstruction (Seo et al., 2023a; Xiao et al., 2022), and leveraging in-the-wild internet-scale datasets (Ma et al., 2022; Ghosh et al., 2023). While these methods are effective and widely utilized, they typically treat visual inputs as 2D grids, overlooking the structured 3D geometric information inherently present in the 3D world. Such a lack of 3D awareness forces the embodied agent to rely on view-specific features such as surface-level pixel patterns or 2D shapes that are unique to the particular perspective, hindering its ability to adapt to different viewpoints or address occlusion. Furthermore, this limitation compels the policy network to implicitly infer 3D actions from 2D visual inputs (2D-to-3D mapping), rather than leveraging 3D-aware representations that are better suited for mapping directly to 3D actions (3D-to-3D mapping). Therefore, learning 3D-aware representations from 2D image inputs is crucial for achieving superior task performance, particularly when precise 3D spatial information inference is critical.

Recent approaches have attempted to learn 3D representation (Ke et al., 2024; Gervet et al., 2023; Goyal et al., 2024). However, these methods typically require not only expert demonstration data but also calibrated cameras for accurate depth projection during deployment. Other prior works have

Figure 1: During pre-training, SinCro learns a view-invariant 3D scene encoder by leveraging cross-view completion with a few randomly selected reference images from different viewpoints via NeRF. During deployment, it utilizes the frozen 3D scene encoder for downstream RL, relying solely on single-view RGB images without performing volume rendering via NeRF.

explored different approaches (Driess et al., 2022; Shim et al., 2023; Li et al., 2022), by mapping multi-view images into a single latent feature and providing it with a volume rendering network in neural radiance field (NeRF) (Mildenhall et al., 2021) to reconstruct the 3D world. Despite these advancements, they often rely on the object-level mask or still require synchronized multi-view images along with camera pose information even during downstream RL. All of these constraints can be a significant burden in real-world robotics applications, where calibration and multi-view setups are impractical. To overcome these challenges, a single-view 3D-aware representation inference framework that relies solely on RGB images is necessary. It is particularly beneficial in practical situations where multiple calibrated cameras are available during pre-training, but the robot has to rely on just single-view RGB images without a camera pose to perform the task during downstream RL deployment.

To develop such an effective 3D-aware representation, we propose a **Sin**gle-view 3D-aware representation inference framework for RL by performing **Cro**ss-view completion via NeRF (**SinCro**). Specifically, it adopts an autoencoder structure, trained only with RGB supervision, and consists of two stages: (1) pre-training a masked ViT-based (Dosovitskiy, 2020) 3D scene encoder through a latent-conditioned NeRF decoder, and (2) deploying only the pre-trained 3D scene encoder in downstream RL tasks. The encoder utilizes a pixel masking strategy (He et al., 2022) with cross-view completion (Weinzaepfel et al., 2022) for 3D geometry-aware representation, and the NeRF decoder leverages multi-view reconstruction with a custom ray sampling strategy to capture inherent, essential 3D information of the environment and fine-grained details crucial for downstream robotic tasks. Additionally, we further regularize the intermediate representation to ensure consistency across different viewpoints by applying a time contrastive loss (Sermanet et al., 2018). Combining all of the proposed components for enhanced 3D-aware representation, our method outperforms the prior works in downstream RL. Further ablation studies and analyses validate that the proposed method is crucial to perform the single-view inference successfully and enables us to obtain representations that are view-invariant and robust to the viewpoint changes.

In summary, this work has the following key contributions:

- We present SinCro, a 3D-aware representation-based RL framework. It can extract 3D-aware representation only with single-view RGB images during the downstream RL.
- The proposed method learns 3D geometry-aware and view-invariant representations of the scene by leveraging a NeRF-based cross-view completion and contrastive learning.
- The proposed method achieves superior downstream RL results, and we qualitatively demonstrate that the learned representation provides an implicit understanding of the 3D world.

## 2 RELATED WORKS

### 2.1 2D REPRESENTATION LEARNING FOR RL

Prior works have been proposed to develop an efficient, effective representation learning strategy for RL in the image domain. Some prior approaches formulate representation learning as encoder

pre-training via auxiliary learning objectives (Yarats et al., 2021b; Liu & Abbeel, 2021a;b), self-supervised reconstruction (Nair et al., 2018; Seo et al., 2023a; Xiao et al., 2022), masked image modeling (Seo et al., 2023a; Xiao et al., 2022; Seo et al., 2023b), and contrastive learning (Sermanet et al., 2018; Laskin et al., 2020; Nair et al., 2022). Other approaches have introduced objectives specialized for decision-making such as predicting future states from the current state (Seo et al., 2022; Hafner et al., 2019; 2020; Hansen et al., 2023), training value functions (Ma et al., 2022; Ghosh et al., 2023), or data augmentations (Yarats et al., 2021a;c). However, these works do not consider the innate 3D structure of the environment, which leads the network to lack 3D geometry awareness and depend on implicit 2D-to-3D mapping. In this work, we utilize the NeRF-based 3D scene representation learning to enhance the 3D understanding of the image feature.

## 2.2 3D Scene Representation

Building on recent progress in robot learning and computer vision, several approaches have emerged that leverage multiple cameras to capture multi-view images for vision-based control (Sermanet et al., 2018; Chen et al., 2021; Hsu et al., 2022; Guhur et al., 2023; Jangir et al., 2022). While most of them directly utilize multi-view images as inputs, they do not perform explicit reconstruction of the 3D world, which is crucial for 3D understanding. Some prior works have attempted to explicitly model the 3D space (Ke et al., 2024; Shridhar et al., 2023; Gervet et al., 2023; Goyal et al., 2024; Qian et al., 2024; Ze et al., 2024), but they require calibrated cameras to get depth images and project the queried pixel into a 3D space during deployment. Other prior works propose to learn implicit 3D-aware representation (Driess et al., 2022; Shim et al., 2023; Li et al., 2022) by reconstructing the 3D world via neural radiance field (NeRF) (Mildenhall et al., 2021). However, multiple cameras are still required in these methods to infer the 3D-aware representation and downstream behavioral learning. Furthermore, some of them even require semantic masks for all pixels in the image for semantic (Shim et al., 2023) or object-level reconstruction (Driess et al., 2022). In this work, we propose a NeRF-based cross-view completion for 3D scene representation learning, trained only with RGB supervision. It allows the inference of 3D scene representation using just single-view images, eliminating the need for calibrated cameras during the downstream RL.

## 3 Preliminary

### 3.1 Visual Reinforcement Learning

In visual RL, we assume a Partially Observable Markov Decision Process (POMDP) $\mathcal{M} = (\mathcal{S}, \mathcal{O}, \mathcal{A}, \mathcal{P}, r, \gamma)$, where $\mathcal{S}$ is the state space of the underlying system, $\mathcal{O}$ is the image observation space, $\mathcal{A}$ is the action space, $\mathcal{P} : \mathcal{S} \times \mathcal{A} \to \Delta(\mathcal{S})$ is the environment dynamics, $r$ is the reward function, and $\gamma$ is the discount factor. The RL objective is to discover a policy $\pi : \mathcal{O} \to \Delta(\mathcal{A})$ that maximizes the expected return $\mathbb{E}_{\pi, \mathcal{P}} \left[ \sum_{t=0}^{\infty} \gamma^t r_t \right]$. Since we utilize the 3D scene encoder $\Omega : \mathcal{O} \to \mathcal{Z}$ that maps the image observation into the latent representation, the RL-relevant networks such as the policy and critic are modified to take $\Omega(\mathcal{O})$ instead of raw image observation $\mathcal{O}$.

### 3.2 Neural Radiance Fields

The idea behind the neural radiance fields (NeRF) (Mildenhall et al., 2021) is to model a 3D scene by predicting a learnable continuous volumetric radiance field. It is represented by differentiable rendering function $F_\theta$ that maps a 3D location $\mathbf{x}$ and viewing direction $\mathbf{d}$ to a color $\mathbf{c}$ and a density $\sigma$, i.e. $F_\theta(\mathbf{x}, \mathbf{d}) = (\mathbf{c}, \sigma)$. To render an image from a specific viewpoint, NeRF aggregates the color information of a camera ray $\mathbf{r}(t) = \mathbf{o} + t\mathbf{d}$, and computes the expected color $C(\mathbf{r})$ as follows:

$$C(\mathbf{r}) = \int_{t_n}^{t_f} T(t)\sigma(\mathbf{r}(t))\mathbf{c}(\mathbf{r}(t), \mathbf{d})dt, \text{ where } T(t) = \exp\left(-\int_{t_n}^{t} \sigma(\mathbf{r}(s))ds\right). \quad (1)$$

where $\mathbf{o}$ is the camera center, $T(t)$ is the accumulated transmittance, and $t_n$, $t_f$ are pre-defined near and far depth bounds, respectively. Then, $F_\theta$ is optimized by pixel-level RGB supervision

$$\mathcal{L}_{RGB} = \sum_{\mathbf{r}_{i,j}} \|\hat{C}(\mathbf{r}_{i,j}) - C(\mathbf{r}_{i,j})\|_2^2 \quad (2)$$

where $\mathbf{r}_{i,j}$ denotes the sampled ray $j$ from camera view $i$. Even though NeRF shows impressive results in 3D scene reconstruction, its key limitation is the assumption of a static scene. Some prior works propose methodologies to model the dynamic scenes (Cao & Johnson, 2023; Park et al., 2023; Pumarola et al., 2021; Fridovich-Keil et al., 2023), but they usually assume a single video input. In other words, the scene should be uniquely determined given a specific timestep $t$. However, in the context of RL, the scene at a specific timestep $t$ could differ across every episode, making the prior works unavailable. To address this issue, we extract essential information from a few images of the current scene and use it as a latent condition for the NeRF model to reconstruct the scene.

## 4 METHOD

This section demonstrates the details of our method. It consists of two stages, pre-training NeRF for representation learning and downstream RL. In section 4.1, we propose a latent-conditioned NeRF model that learns 3D geometry-aware scene representations. In section 4.2, time contrastive loss is proposed to regularize the representation. In section 4.3, we introduce an RL algorithm that leverages the 3D aware representation extracted from the pre-trained encoder with single-view input.

### 4.1 3D-AWARE REPRESENTATION LEARNING WITH CROSS-VIEW COMPLETION VIA NERF

To obtain 3D-aware scene representations and address the dynamic scenes, we learn a 3D scene encoder $\Omega_\theta$ that maps the image observations to a latent scene representation $z$ for each timestep and learns the rendering function $F_\theta$ based on the latent $z$, i.e. $F_\theta(\mathbf{x}, \mathbf{d}, z)$. Specifically, we employ a pretext task of cross-view completion (Weinzaepfel et al., 2022). It involves reconstructing an input image with masked sections by utilizing the visible content and referring to unmasked reference images from different viewpoints.

**Overview.** The overall framework is shown in Figure 2. Assuming access to a dataset consisting of a few episodic rollout videos captured from $N$ different viewpoints, let $O_{t-2:t}^i$ denote the primary image observations taken from the $i^{th}$ viewpoint at timestep $t - 2 : t$. We concatenate three consecutive images to incorporate trajectory history into the latent scene representations. Since we leverage reference images for cross-view completion tasks, we also denote $O_{t-2:t}^{r_j}$ as the reference image observations taken from the $j^{th}$ viewpoint at timestep $t - 2 : t$.

For every training iteration, we randomly select primary images from a specific viewpoint ($O_{t-2:t}^i$) and $K$ different reference images from other viewpoints except the viewpoint of the primary image ($O_{t-2:t}^{r_1}, \cdots, O_{t-2:t}^{r_K}$). All of the selected images are divided into non-overlapping patches $P_{t-2:t}^i, P_{t-2:t}^{r_1}, \cdots, P_{t-2:t}^{r_K}$, and we mask randomly selected patches from $P_{t-2:t}^i$ with masking ratio $m$, denoted as $P_{m,t-2:t}^i$. The masking is applied to ensure the 3D scene encoder $\Omega_\theta$ learns both 3D geometry-aware information by cross-view completion and contexts within the masked viewpoint by restoring the original images. Each of $P_{m,t-2:t}^i, P_{t-2:t}^{r_1}, \cdots, P_{t-2:t}^{r_K}$ is independently passed through a shared ViT-based image encoder $\mathcal{E}_\theta$. Then, the outputs are concatenated and passed through a ViT-based state encoder $\mathcal{S}_\theta$, resulting in state features corresponding to each image at the latest timestep, $v_t^i, v_t^{r_1}, \cdots, v_t^{r_K}$. Finally, these features are combined to generate the latent scene representation $z_t$ and the rendering function $F_\theta$ synthesizes images from multiple viewpoints conditioned on $z_t$.

**Details on the image encoder.** The shared image encoder $\mathcal{E}_\theta$ follows the standard ViT structure, but it is modified to meet the requirement for the downstream RL. Specifically, to deal with the consecutive images, we add 1D learnable parameters representing each timestep along with 2D sinusoidal positional embeddings for the patches. Then, the patches $P_{m,t-2:t}^i, P_{t-2:t}^{r_1}, \cdots, P_{t-2:t}^{r_K}$ are independently passed through transformer blocks in ViT.

**Details on the state encoder.** The ViT-based state encoder $\mathcal{S}_\theta$ follows a similar structure to the image encoder. It concatenates the encoded patches from the image encoder to facilitate the exchange of scene information across views. Specifically, for the primary images, we take the encoded patches across all timesteps to guide the state encoder's attention toward the temporal information which is crucial for downstream RL such as the agent's movement or interaction with an object. For the reference images, we only take the encoded patches of the latest timestep $t$ to encourage the state

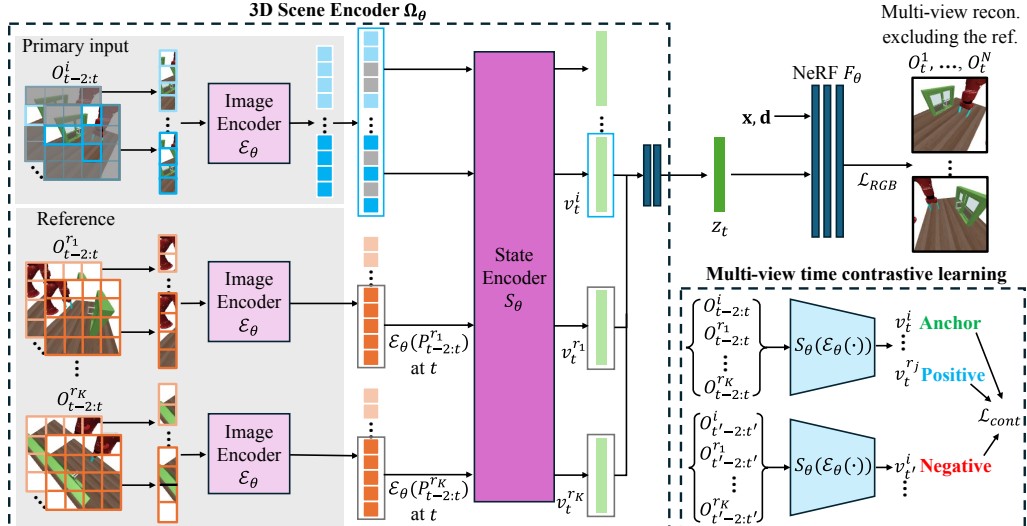

Figure 2: 3D scene encoder $\Omega_\theta$ takes masked primary images and $K$ reference images as inputs to extract the latent scene representation $z_t$. It is trained by cross-view completion via NeRF, and time contrastive learning is applied to regularize the scene representation to be view-invariant. Once the pre-training is finished, only $\Omega_\theta$ will be used for downstream RL, and NeRF will no longer be used.

encoder to focus on 3D geometric information from different viewpoints, while improving memory and computation efficiency. To fill out the missing patches corresponding to the masked ones, learnable mask tokens are concatenated (He et al., 2022). We add 1D learnable parameters representing whether the encoded patches are from primary images or reference images, along with 1D learnable parameters representing each timestep and 2D sinusoidal positional embeddings for the patches.

After processing the concatenated inputs by $\mathcal{S}_\theta$, the encoded output chunks corresponding to each $\mathcal{E}_\theta(P^i_{m,t-2:t}), \mathcal{E}_\theta(P^{r_1}_{t-2:t}), \cdots, \mathcal{E}_\theta(P^{r_K}_{t-2:t})$ at the latest timestep $t$ are denoted as $v^i_t, v^{r_1}_t, ..., v^{r_K}_t$, called state features. Then, the latent scene representation $z_t$ is computed by averaging these state features, followed by a shallow MLP projection and L2 normalization. For notational simplicity in downstream RL, we define $z_t = \Omega_\theta(O^i_{t-2:t}, O^{r_1}_{t-2:t}, \cdots, O^{r_K}_{t-2:t})$, which includes all the processes from the observation images to the latent scene representation.

**NeRF decoder for multi-view reconstruction.** Leveraging 2D images from different viewpoints, the rendering function $F_\theta$ is trained to reconstruct images conditioned on the latent scene representation $z_t$. Since NeRF is inherently designed to model the 3D scene, it naturally encourages the 3D structural understanding of the 3D scene encoder compared to the typical 2D convolutional neural network-based one. Also, to further enhance the 3D awareness of $z_t$, we additionally employ multiview reconstruction, unlike the prior self-supervised works with masked modeling (He et al., 2022; Weinzaepfel et al., 2022). Specifically, we reconstruct both the masked primary image and images from all other viewpoints at the latest timestep $t$, while excluding the reference images to prevent self-reconstruction, which would bypass proper 3D understanding of the environment. As the 3D scene encoder $\Omega_\theta$ learns not just the cross-view information from the reference images, but also the information from other viewpoints not included in the 3D scene encoder's inputs, this approach encourages $\Omega_\theta$ to capture the essential 3D information of the environment.

Also, we have found that NeRF training often gets stuck in local minimum since the NeRF model tries to reconstruct every single pixel even though it corresponds to non-salient parts such as background or static objects. It leads to degradation in capturing the fine-grained details crucial for downstream tasks. To address this, we introduce object-focused ray sampling to improve the fine-grained details of the object-of-interest in rendered images. While there are prior works that utilize adaptive sampling (Lin et al., 2022; Rematas et al., 2022), or surface, depth, pixel value changes (Piala & Clark, 2021; Sun et al., 2024), we uniquely propose to sample rays for the specific regions, potentially crucial to downstream tasks. It involves weighted sampling of rays within the region of interest, identified using the Grounded SAM (Ren et al., 2024), instead of uniform random sampling. By adjusting ray sampling locations, our method strikes a balance between focusing on non-salient parts and regions crucial for downstream tasks. More details are included in Appendix A.

## 4.2 REGULARIZATION FOR VIEWPOINT-INVARIANCE

In addition to the cross-view completion, we propose to regularize $z_t$ by applying a multi-view time contrastive loss (Sermanet et al., 2018) to the state features $v_t^i, v_t^{r_1}, ..., v_t^{r_K}$ to ensure the 3D scene encoder $\Omega_\theta$ is view-invariant. Specifically, the multi-view time contrastive loss encourages a pair of simultaneously observed state features from different viewpoints to be closer to each other, while repulsing state features from the same viewpoint but different timesteps. We set $v_t^i$ as an anchor and randomly select a state feature from $v_t^{r_j}$, where $j \in \{1, \cdots, K\}$, as a positive, and set $v_{t'}^i$, where $t'$ indicates a timestep distant from $t$, as a negative. Then, the objective can be represented as follows:

$$\mathcal{L}_{\text{cont}} = \max \left( \left\| \boldsymbol{v}_t^i - \boldsymbol{v}_t^{r_j} \right\|_2^2 - \left\| \boldsymbol{v}_t^i - \boldsymbol{v}_{t'}^i \right\|_2^2 + \alpha, 0 \right) \qquad (3)$$

where $\alpha$ is the margin that encourages dissimilar pairs and positive pairs to be distant. Finally, the overall loss function is formulated as

$$\mathcal{L}_{total} = \mathcal{L}_{RGB} + \lambda_{cont}\mathcal{L}_{\text{cont}} \qquad (4)$$

where $\lambda_{cont} = 0.0004$. Since we have encouraged the 3D scene encoder $\Omega_\theta$ to be not only 3D geometry-aware but also view-invariant by leveraging the cross-view completion via NeRF along with the multi-view time contrastive loss, $\Omega_\theta$ is capable of performing inference solely with single-view images, which is practically desirable for the downstream robotic tasks via RL.

## 4.3 REINFORCEMENT LEARNING WITH 3D-AWARE REPRESENTATION

Once we train the 3D scene encoder $\Omega_\theta$, it is exploited as a 3D-aware representation extractor for the downstream RL algorithm with single-view input. The 3D scene encoder $\Omega_\theta$ takes $K$ times replicated $O_{t-2:t}^i$ instead of using reference images from different viewpoints, i.e. $z_t = \Omega_\theta(O_{t-2:t}^i, [O_{t-2:t}^i] * K)$, where $*$ denotes replication. Since we consider a deployment setting with observation images from a single viewpoint, we randomly select a viewpoint from those in the NeRF pre-training dataset for each episode during the RL process, instead of capturing synchronized multi-view images. Then, the RL-relevant networks such as the policy and critic take $z_t$ as an input observation. During the downstream RL process, we do not apply masking and freeze the 3D scene encoder's weight to preserve the 3D scene representation.

In this work, we adopt DrM (Xu et al., 2023) for the downstream RL algorithm, which is built on top of DrQ-v2 (Yarats et al., 2021a). It utilizes the dormant ratio of the neural network for active exploration-exploitation scheduling and shows state-of-the-art performance in the visual RL domain.

## 5 EXPERIMENT

Following the prior work (Shim et al., 2023), we pre-train and evaluate the proposed method for each environment in the Meta-world (Yu et al., 2020), where the robot should perform manipulation tasks with randomly initialized object-of-interest. However, we modified the environment to make it look more realistic and rich in texture compared to the prior work. For the dataset, we recorded episodic videos from six distinct viewpoints ($N = 6$). We used two reference images ($K = 2$) during NeRF pre-training. However, with additional computational resources, both $N$ and $K$ could be expanded. More details about the environments and datasets are included in Appendix A.

To evaluate the learned representation's effectiveness in downstream RL, we compare our method with some prior 3D-aware representation-based RL methods, which can be summarized as follows:

**NeRF-RL** (Driess et al., 2022) – it performs NeRF-based 3D reconstruction by leveraging object masks for object-level reconstruction, which are required both in pre-training and deployment.

**SNeRL** (Shim et al., 2023) – it performs NeRF-based 3D reconstruction while distilling the feature field of DINO (Caron et al., 2021) and semantic labels of each pixel into the latent representation.

**3D-NSR** (Li et al., 2022) – it learns **3D N**eural **S**cene **R**epresentations by performing self-reconstruction of multi-view images via NeRF while enforcing time contrastive loss for view-invariancy. SNeRL, NeRF-RL, and 3D-NSR require camera pose information and synchronized images from multiple viewpoints during the downstream RL process.

Table 1: Conceptual comparison between the proposed method and other baselines. **Single-view**: whether the 3D-aware representation can be inferred with single-view input during deployment. **Supervision source**: the external supervision source of the representation learning objective. **Without calibrated cameras**: the requirement for camera viewpoint information during deployment. **Reconstruction**: whether the method performs self-reconstruction or cross-view reconstruction.

| | Pre-training | | Downstream RL deployment | |
| --- | --- | --- | --- | --- |
| | Reconstruction | Supervision source | Without calibrated cameras | Single-view |
| SNeRL | Self | RGB, Feature Field, Semantic Label | ✗ | ✗ |
| NeRF-RL | Self | RGB, Object mask | ✗ | ✗ |
| 3D-NSR | Self | RGB | ✗ | ✗ |
| **SinCro(ours)** | Cross-view | RGB | ✓ | ✓ |

**CNN+view randomization** – as a 2D representation baseline, it constructs a naive 2D CNN as an encoder for the downstream RL, without using the 3D scene encoder, NeRF-based 3D reconstruction. We apply random cropping for data augmentation to align with the recent image-based RL training recipe. It performs RL with single-view input while randomly selecting the viewpoint for every episode, the same as SinCro is trained.

For all baselines, we concatenate proprioceptive states such as the XYZ position of the end-effector with the learned representation from each method to account for the robot's internal state, and use this combined data as input for the RL agent. While proprioceptive data is inherently view-agnostic, it serves a complementary role, focusing solely on the robot's dynamics. On the other hand, the learned 3D-aware representation remains crucial for understanding and interacting with the environment and object-of-interest, especially in tasks requiring spatial awareness and manipulation. By combining these two, we ensure that our method is broadly applicable without imposing restrictive assumptions on the robotics problem and compromising the importance of the 3D-aware representation. We utilize DrM as the downstream RL algorithm with the same training process for a fair comparison. The conceptual comparison of the baselines and SinCro is shown in Table 1.

## 5.1 RL EXPERIMENT RESULTS & 3D SCENE REPRESENTATION VALIDATION

**RL experiments.** The evaluation results for downstream RL are shown in Figure 3. Since our method and CNN+view randomization utilize only single-view input, we evaluate these with each viewpoint used in the NeRF pre-training phase and average the results from all viewpoints. Compared to other baselines that utilize multi-view images to infer 3D-aware representation such as SNeRL, NeRF-RL, and 3D-NSR, the proposed method consistently shows superior downstream RL performances despite using single-view input. Since the proposed method mostly outperforms these baselines rather than just being comparable, it supports the significance of the proposed architecture and training framework for 3D geometry-aware representation. In the case of CNN+view randomization, it shows performance degradation compared to our method since it has to learn every single representation from each different viewpoint due to the lack of 3D awareness. It shows that the proposed 3D geometry-aware, view-invariant representation is crucial for consistent, reliable downstream RL performances.

**3D scene representation.** To validate whether the proposed method can effectively extract the essential information required to represent the 3D scene, we compare the 3D volume rendering results of the proposed method and baselines. Note that these qualitative results are indirect validations for 3D understanding of our learned latent representation $z_t$, and the high-quality image synthesis is not the primary objective. Given a task-solving trajectory that was not used in pre-training, the evaluation is performed with a single-view input.

As shown in Figure 4, the proposed method demonstrates superior scene-representing capabilities. SinCro successfully reconstructs the core components of the scene when only a single-view image is provided. It consistently achieves superior quantitative results compared to other baselines. Also, it can accurately position the box for peg insertion, whereas other baselines often overfit to incorrect positions. This distinction becomes particularly evident when rendering from different viewpoints (V1, V2). This represents that the proposed method implicitly captures the spatial information of the 3D world. Additionally, other baselines struggle to render key elements such as the robot arm and peg, often exhibiting issues such as jittering, and teleportation. We also compare with the baselines in the multi-view input setting. The results for this setting are included in the Appendix B. Consid-

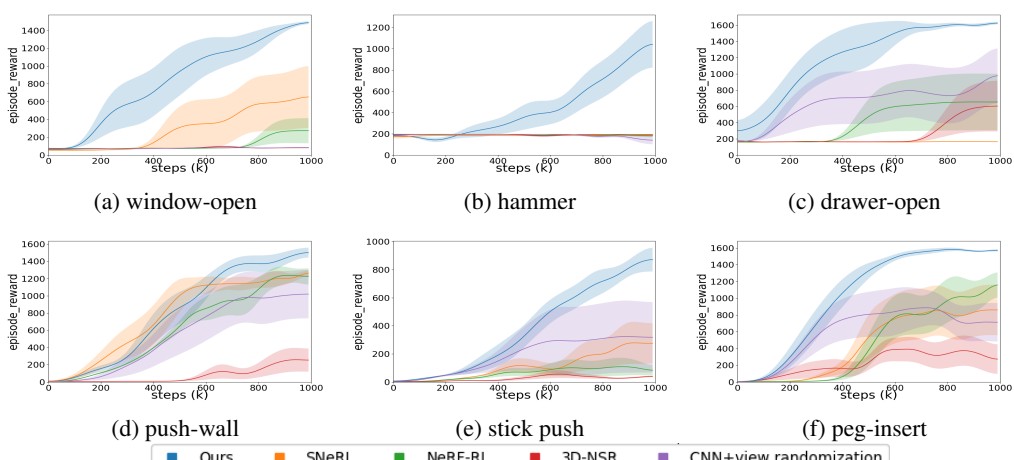

(a) window-open       (b) hammer       (c) drawer-open

(d) push-wall       (e) stick push       (f) peg-insert

■ Ours    ■ SNeRL    ■ NeRF-RL    ■ 3D-NSR    ■ CNN+view randomization

Figure 3: Comparison of our method and baselines in Meta-world environments. The shaded regions represent a standard deviation across different seeds. Note that most of the NeRF-based baselines fail to perform the task with single-view input, so we do not include the results of these.

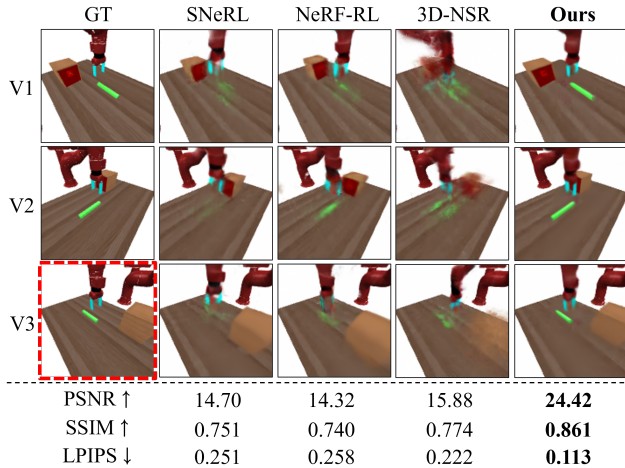

| | SNeRL | NeRF-RL | 3D-NSR | **Ours** |
|---|---|---|---|---|
| PSNR ↑ | 14.70 | 14.32 | 15.88 | **24.42** |
| SSIM ↑ | 0.751 | 0.740 | 0.774 | **0.861** |
| LPIPS ↓ | 0.251 | 0.258 | 0.222 | **0.113** |

Figure 4: 3D volume rendering results of the peg-insert environment (best viewed in the digital version). All methods take single-view input from V3 (outlined in red). SinCro demonstrates its ability to accurately localize the object-of-interest in the scene and achieves more consistent quantitative results than baselines. Note that we only visualize three among all viewpoints due to the page limit.

ering these baselines exploit synchronized multi-view input or additional supervision sources, the superior 3D spatial awareness of SinCro demonstrates the effectiveness of the proposed framework in capturing detailed 3D dynamic scenes, even with just RGB supervision and single-view inference.

## 5.2 VIEWPOINT-INVARIANCE

**RL experiments.** Considering the real-world application, it would be beneficial to conduct downstream RL with minimal randomization in viewpoints across episodes. To validate such property, we perform the same RL experiments in Section 5.1, while utilizing only one or two viewpoints for random view selection during the RL process. We expect that if the learned representation is view-invariant, the policy will be able to sample an action similar to what would have been chosen from the viewpoint used during the RL process, even when the inputs for $\Omega_\theta$ come from a previously unseen viewpoint during the RL process. As shown in Figure 5, the performance is quite robust to the decreases in the number of viewpoints, indicating that the learned representation is view-invariant.

**3D scene representation.** To validate whether the latent scene representation $z_t$ is viewpoint-invariant, we plot t-SNE embeddings of $z_t$ in Figure 5. We collect videos of a task-solving trajectory from six different viewpoints and infer $z_t$ for all timesteps and viewpoints with 1) multi-view input,

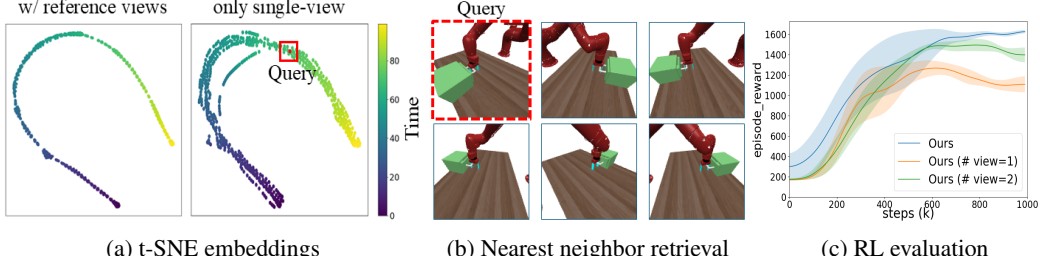

| (a) t-SNE embeddings | (b) Nearest neighbor retrieval | (c) RL evaluation |

Figure 5: Viewpoint-invariance analysis in drawer-open environment. (a) t-SNE embeddings of videos from six distinct viewpoints are aligned with similar timesteps across different viewpoints, even when extracted without reference images. (b) The nearest neighbor search for the query image (outlined in red) retrieves temporally aligned images from each viewpoint. (c) RL evaluation results with different numbers of viewpoints for random view selection. These results demonstrate the view-invariant properties of the representations.

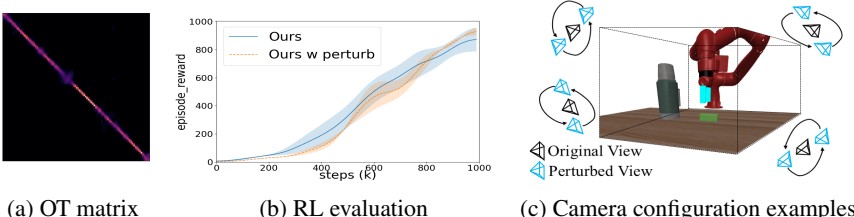

| (a) OT matrix | (b) RL evaluation | (c) Camera configuration examples |

Figure 6: Viewpoint perturbation analysis in stick-push. (a) Optimal transport (OT) matrix between episodic task-solving videos captured from the default and perturbed viewpoints. (b) RL evaluation results with/without the perturbations. (c) Camera configurations for viewpoint perturbations.

and 2) single-view input. The results demonstrate locally smooth and clear temporal progress of all trajectories, while the representations are closely aligned with similar timesteps across different viewpoints, even with single-view input. To further validate the view-invariance, we randomly select an image from arbitrary timestep and viewpoint, then retrieve the nearest neighbors in the latent scene representation space based on the Euclidean distance metric. The retrieved images from each viewpoint are temporally aligned, highlighting that the latent scene representation remains consistent regardless of the viewpoints.

### 5.3 ROBUSTNESS TO VIEWPOINT CHANGES

To further explore the benefits of 3D geometry-aware representation, we consider a viewpoint-robust control setup by continuously shifting the camera to be centered around the default pose, adjusting it by approximately five degrees in both azimuth and elevation. We evaluate the learned RL policy in Section 5.1 under these camera perturbations. As shown in Figure 6, our method is robust to the viewpoint changes (**Ours w perturb**) due to the 3D geometry-aware representation. We also compute the optimal transport matrix (Haldar et al., 2023; Hu et al., 2023) for the single-view-based latent scene representations between episodic task-solving videos captured from the default and perturbed viewpoints. It represents the alignment between two distributions, so if the representation is robust to viewpoint changes, we would expect to observe high values along the diagonal, indicating strong correspondence. We notice that high values remain concentrated on the diagonal, even though these perturbed viewpoints were not provided during the NeRF pre-training. It represents that the latent scene representation maintains strong local consistency and is resilient to variations in camera configurations, offering a practical advantage for deployment in downstream robotic tasks.

### 5.4 ABLATION STUDY

To investigate the contribution of each proposed component, we compare the single-view-based rendering results and downstream RL performance by removing each one (Figure 7).

**Cross-view completion and contrastive learning** – we ablate reference images (**w/o cross-view**) or objective function (3) (**w/o contrastive**) during pre-training. Without one of these, the RL agent always fails to perform the task, so we do not include the results. As shown in Figure 7, the absence

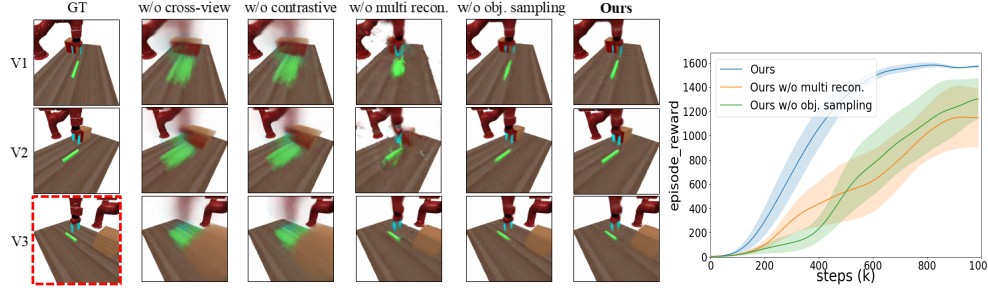

(a) Visual comparison for ablation study   (b) RL evaluation

Figure 7: Ablation study in peg-insert environment (best viewed in the digital version). (a) Rendering results of single-view inference with a primary input V3 (outlined in red). Note that we only visualize three viewpoints in the dataset due to the page limit. (b) RL evaluation by ablating multi-view reconstruction and object-focused ray sampling.

of these components leads to collapsed reconstructions, appearing as blurred images. It indicates the importance of both cross-view completion, which enhances the 3D scene encoder's understanding of the 3D geometry of the environment, and contrastive learning, which offers crucial guidance in distinguishing different scenes. Together, they provide complementary benefits during pre-training, ensuring that the 3D-aware representation integrates both spatial and temporal aspects, ultimately enabling the RL agent to accomplish the task.

**Multi-view reconstruction** – the absence of multi-view reconstruction (**w/o multi recon.**) leads to RL performance degradation. This is because the 3D scene encoder appears to lose some 3D scene awareness, which is essential for effective policy learning. For example, the failure to localize key elements such as the robot gripper, unobserved parts in the primary image, and the accurate position of the green peg impacts the downstream RL tasks. It suggests that multi-view reconstruction plays a critical role in enhancing the encoder's understanding of 3D geometry, eventually enhancing the downstream RL performance.

**Object-focused ray sampling** – there is also a noticeable degradation in RL performance when the object-focused ray sampling strategy (**w/o obj. sampling.**) is not applied. This is because the blurry or incomplete reconstructions near small objects, such as the green peg, which are often accompanied by jittering across viewpoints, hinder the 3D scene encoder's ability to accurately localize the object. Accurate object localization is critical for solving the robotic manipulation task, directly impacting the effectiveness of the downstream RL.

More experimental results, analysis, and evaluation details in Section 5 are included in supplementary video and Appendix B

## 6 CONCLUSION

In this work, we considered a 3D-aware representation-based RL framework where the agent should learn how to perform the given task from multiple viewpoints. We proposed SinCro, which can extract 3D geometry-aware representation while enabling a single-view inference without synchronized calibrated cameras during deployment. We have shown that the proposed method outperforms the baselines in qualitative and quantitative ways. However, SinCro still has some limitations. For example, we have experimented with a small number of camera viewpoints due to the huge requirements of the computational resources, but access to more camera viewpoints will enable us to obtain more fine-grained details and perform more viewpoint-robust control. Also, the proposed method still requires synchronized and calibrated cameras during the NeRF pre-training, which might hinder real-world application if we want it to be used with dozens of cameras. An interesting future research direction could involve using multiple videos captured from moving cameras with varying viewpoints in place of the current prerequisites.

**Reproducibility Statement.** We have included the source code in the supplementary materials, along with detailed instructions for running the experiments. All necessary configurations, datasets, and parameters used in our experiments are clearly specified, ensuring that others can replicate our results under the same conditions and achieve consistent outcomes.

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

## A  EXPERIMENTAL DETAILS

### A.1  ENVIRONMENT

- **window-open**: The window's spatial location is randomly initialized on the table with the window in a closed position. The robot should open the window by manipulating the handle.

- **drawer-open**: The drawer's spatial location is randomly initialized on the table with the drawer in a closed position. The robot should open the drawer by manipulating the handle.

- **hammer**: The hammer's spatial location is randomly initialized on the table. The robot should manipulate the hammer to hit the nail attached to the box.

- **peg-insert-side**: The peg and box's spatial location is randomly initialized on the table. The robot should pick up the peg and insert it into the hole in the box.

- **push-wall**: The block's spatial location is randomly initialized on the table. The robot should push the block to the desired location while detouring the wall.

- **stick-push**: The stick's spatial location is randomly initialized on the table. The robot should pick up the stick and push the bottle to the desired location by using the stick.

### A.2  SINCRO IMPLEMENTATIONS

We used NVIDIA A6000 and AMD EPYC 9274F for encoder pre-training, and NVIDIA A5000 and AMD Ryzen Threadripper 3960X for RL training. Each encoder pre-training took 1 day and each RL experiment took 2∼3 days for RL training.

**Dataset for pre-training NeRF.**  Following the prior work (Shim et al., 2023), the dataset for each environment consists of 14400 scenes (120 episodes x 120 timesteps). Half of the dataset is collected by using the scripted policy (suboptimal) provided by Meta-world (Yu et al., 2020) and the other half is collected by random action. We record images from 6 different viewpoints located in the hemisphere with a radius of 0.6 m, centered around the environment. All cameras are set up to look at the center of the hemisphere. The size of observed images from each camera is $128 \times 128$.

**Encoder pre-training details in Section 4.**  The hyperparameters for encoder pre-training are presented in Table 2. We conducted the encoder pre-training for a total of 300K iterations with a batch size of 8. Each data in the batch includes a pair of images (e.g. one for the primary image, two for the reference images, and four for the reconstruction) from a specific episode and timestep. During the first 2000 training iterations, we sampled rays from center-cropped image regions with size $64 \times 64$ to facilitate initial training by focusing on rich texture regions.

For each training iteration, we randomly sample an episode, a timestep, and three non-duplicated camera viewpoints from a dataset, then concatenate three temporally consecutive images from each viewpoint. One of these images serves as the primary input ($O_{t-2:t}^{i}$) and the remaining two viewpoints serve as references ($O_{t-2:t}^{r_1}, O_{t-2:t}^{r_2}$). The image patches ($P_{t-2:t}^{i}, P_{t-2:t}^{r_1}, P_{t-2:t}^{r_2}$) are embedded into 256 dimensions before entering the image encoder. Each ViT block in the image encoder $\mathcal{E}_\theta$ consists of multi-head self-attention and an MLP layer, and then layer normalization is applied at the end of the image encoder.

Both the primary input and reference patches are processed by a fully connected layer, before concatenating mask tokens to the primary input patches from the image encoder. Then, the concatenated inputs and reference patches are fed into transformer blocks in the state encoder $\mathcal{S}_\theta$, followed by layer normalization for each view. To obtain the state features for the primary input and references, $v_t^i, v_t^{r_1}, v_t^{r_2}$, the latest timestep patches for each view are concatenated and embedded into 256 dimensions via a fully connected layer. The scene representation $z_t$ is subsequently generated by averaging the state features, followed by processing through a two-layer MLP and L2 normalization. Then, a latent-conditioned NeRF $F_\theta(\mathbf{x}, \mathbf{d}, z_t)$ utilizes this scene representation as an additional input. Specifically, building upon the vanilla NeRF architecture (Mildenhall et al., 2021), the scene representation $z_t$ is concatenated with the high-frequency embedded position $\mathbf{x}$ both in the first and fifth MLP layers.

We additionally sample negative image observations at each training iteration for time contrastive learning. These negative observations are acquired from the same viewpoints of the previous input batch but they correspond to a different timestep within the same episode. The negative image observations are processed through the frozen image encoder and state encoder, $\mathcal{S}_\theta(\mathcal{E}_\theta(.))$. The resulting state feature from the primary viewpoint, $v_{t'}^i$, is used as a negative pair for the time contrastive loss $\mathcal{L}_{\text{cont}}$, with a margin of $\alpha = 0.2$.

Additionally, we introduce object-focused ray sampling to improve the fine-grained details of small objects in the rendered images, which is crucial for accomplishing downstream robotic tasks. Specifically, we identified regions of interest by utilizing the off-the-shelf text-prompt-based segmentation network, Grounded SAM (Ren et al., 2024). We use text prompt for each environment as follows: green window, white window handle for the window-open environment, light green drawer, white drawer handle for the drawer-open environment, hammer with green hand, gray nail for the hammer environment, green rectangular stick for the peg-insert-side environment, small green cube for the push-wall environment, and green peg for the stick-push environment. During NeRF pre-training, half of the rays are sampled within the region of interest and the other rays are sampled uniformly from the entire image pixels.

Table 2: Hyperparameters for pre-training

| | | | |
|---|---|---|---|
| # of viewpoints, $N$ | 6 | non-linearity | ReLU |
| # of references, $K$ | 2 | optimizer | AdamW |
| masking ratio, $m$ | 75 % | learning rate | 5e-4 |
| image resolution | $128 \times 128$ | patch size | $16 \times 16$ |
| batch size | 8 | $\alpha$ | 0.2 |
| embedding dimensions of $\mathcal{E}_\theta$ and $\mathcal{S}_\theta$ | 256 | hidden units (MLP) in $\mathcal{E}_\theta$ and $\mathcal{S}_\theta$ | 1024 |
| # of transformer blocks in $\mathcal{E}_\theta$ | 4 | # of attention heads in $\mathcal{E}_\theta$ | 4 |
| # of transformer blocks in $\mathcal{S}_\theta$ | 2 | # of attention heads in $\mathcal{S}_\theta$ | 2 |
| MLP layers for NeRF | 8 | hidden units (MLP) in NeRF | 256 |
| # of rays per iteration | 2048 | # of samples per ray | 64 |

**RL experiments details in Section 5.1.** The hyperparameters for downstream RL are shown in Table 3. DrM-specific hyperparameters are adopted from the default values of the original implementation. For proprioceptive inputs, we utilize the end-effector's XYZ position (3-dim) and gripper state that represent its open/close status (1-dim). We additionally utilize a high-frequency mapping function in NeRF Mildenhall et al. (2021) to embed this 4-dimensional proprioceptive state into a high-dimensional feature before passing it to the RL agent.

During RL training, we randomly sampled the camera viewpoint from the dataset for pre-training NeRF. Since there are six distinct camera viewpoints in the dataset, we randomly sample a viewpoint from these six viewpoints at each episode. Then, we evaluate the RL agent with each viewpoint for every 10k steps, compute the episode rewards for each viewpoint, and plot the averaged episode rewards in evaluation graphs in Section 5.

Table 3: Hyperparameters for downstream RL

| | | | |
|---|---|---|---|
| critic hidden dimension | 512 | exploitation expectile | 0.9 |
| critic hidden depth | 2 | exploitation temperature T´ | 0.02 |
| critic target $\tau$ | 0.005 | target exploitation parameter $\hat{\lambda}$ | 0.6 |
| critic target update frequency | 2 | awaken exploration temperature T | 0.1 |
| actor hidden dimension | 512 | linear exploration stddev. clip | 0.3 |
| actor hidden depth | 2 | $n$-step returns | 10 |
| actor update frequency | 2 | $\tau$-dormant ratio | 0.025 |
| batch size | 128 | replay buffer $\mathcal{B}$ capacity (# of transitions) | 5e5 |
| Replay buffer $\mathcal{B}$ capacity (# of transitions) | 5e5 | dormant ratio threshold $\hat{\beta}$ | 0.2 |
| discount factor $\gamma$ | 0.99 | minimum perturb factor $\alpha_{min}$ | 0.2 |
| proprioception layer hidden dimension | 256 | maximum perturb factor $\alpha_{max}$ | 0.9 |
| proprioception layer depth | 2 | perturb interval | 100000 |
| learning rate | 1e-4 | optimizer | ADAM |
| episode steps | 200 | training steps | 1000000 |

**Viewpoint-invariance experiments details in Section 5.2.** **RL** – we train the RL agent, following the same process in Section 5.1. However, we select only a single viewpoint and do not randomize the viewpoint during the RL process (**Ours (# view=1))** in Figure 6), or select two viewpoints and only randomize the viewpoint within these two during the RL process (**Ours (# view=2))** in Figure 6). For example, let's assume there are viewpoints [1,2,3,4,5,6]. If we selected [6] in the case of (# view=1), then we train the RL agent only with the image from viewpoint 6 during the RL process, and evaluate the trained RL agent with all viewpoints [1,2,3,4,5,6]. Ideally, if the learned 3D-aware representation is perfectly view-invariant, the output of the 3D scene encoder will be the same regardless of the viewpoint, leading to sampling the same action regardless of the viewpoint.

**Robustness to viewpoint changes experiments details in Section 5.3.** For perturbation, we continuously shift the camera to follow the circle around the default camera pose, while maintaining the maximum deviation of azimuth and elevation angle as 5 degrees for each. The camera is set to rotate in the circle every 20 steps.

### A.3 BASELINE IMPLEMENTATIONS

The baseline algorithms are trained as follows,

- **SNeRL** (Shim et al., 2023): We refer to the original implementation in `https://github.com/jayLEE0301/snerl_official`, and follow the default setting.
- **NeRF-RL** (Driess et al., 2022): Since there is no official code implementation, we implemented it ourselves by closely following the paper.
- **3D-NSR** (Li et al., 2022): Since there is no official code implementation, we implemented it ourselves by closely following the paper.
- **CNN+view randomization**: We replace the proposed encoder with a CNN and utilize random cropping of the image for data augmentation.

### A.4 ALGORITHM

---

**Algorithm 1** pre-training of SinCro

---

1: **Definition:** total training iterations $N$, multi-view dataset $\mathcal{D}$, 3D scene encoder $\Omega_\theta$, image encoder $\mathcal{E}_\theta$, state encoder $\mathcal{S}_\theta$, NeRF decoder $F_\theta$, $\lambda_{cont} = 0.0004$
2: **for** iteration=1,2,...,$N$ **do**
3:     randomly select the primary input viewpoint $i$ and $K$ reference viewpoints.
4:     sample observations $O_{t-2:t}^i, O_{t-2:t}^{r_1}, \ldots, O_{t-2:t}^{r_K}$ from $\mathcal{D}$
5:     $z_t \leftarrow \Omega_\theta(O_{t-2:t}^i, O_{t-2:t}^{r_1}, \ldots, O_{t-2:t}^{r_K})$
6:     reconstruct $\hat{O}_t^j, \forall j$ via $F_\theta$ conditioned on $z_t$, while excluding the reference viewpoints, and compute $\mathcal{L}_{RGB}$.
7:     sample negative observations $O_{t'-2:t'}^i, O_{t'-2:t'}^{r_1}, \ldots, O_{t'-2:t'}^{r_K}$ within the same episode but difference timestep $t'$.
8:     $v_t^i, v_t^{r_1}, \ldots, v_t^{r_K} \leftarrow \mathcal{S}_\theta(\mathcal{E}_\theta(O_{t-2:t}^i, O_{t-2:t}^{r_1}, \ldots, O_{t-2:t}^{r_K}))$ (already obtained during line 5)
9:     $v_{t'}^i, v_{t'}^{r_1}, \ldots, v_{t'}^{r_K} \leftarrow \mathcal{S}_\theta(\mathcal{E}_\theta(O_{t'-2:t'}^i, O_{t'-2:t'}^{r_1}, \ldots, O_{t'-2:t'}^{r_K}))$
10:     compute $\mathcal{L}_{cont}$ using $v_t^i, v_t^{r_j}$ and $v_{t'}^i$, where $j \in \{1, \cdots, K\}$.
11:     $\mathcal{L}_{total} \leftarrow \mathcal{L}_{RGB} + \lambda_{cont}\mathcal{L}_{cont}$
12:     update the parameters of $\Omega_\theta$ and $F_\theta$ by minimizing $\mathcal{L}_{total}$
13: **end for**

---

---

**Algorithm 2** Downstream RL of SinCro

---

1: **Definition:** environment horizon $H$, total training episodes $N$, env, policy $\pi_\psi$, critic $Q_\psi$, replay buffer $\mathcal{B}$, pre-trained encoder $\Omega_\theta$, number of reference views $K$, proprioceptive state $s$,
2: **for** iteration=1,2,...,$N$ **do**
3:     randomly select viewpoint $i$ from the NeRF pre-training dataset $\mathcal{D}$ in Algorithm 1.
4:     $O_0^i, s_0 \sim$ env.reset()
5:     **for** $t$=0,1,...,$H$-1 **do**
6:         $a_t \leftarrow \pi_\psi(\cdot|\Omega_\theta(O_t^i, [O_t^i] * K), s_t)$
7:         $O_{t+1}^i, s_{t+1} \leftarrow$ env.step($a_t$)
8:     **end for**
9:     $\mathcal{B} \leftarrow \mathcal{B} \cup \{O_0^i, s_0, a_0, O_1^i, s_1, ...\}$
10:    Get a minibatch b from $\mathcal{B}$ and train the policy $\pi_\psi$ and the critic $Q_\psi$ with b via DrM (Xu et al., 2023)
11: **end for**

---

## B   Additional Experimental Results

### B.1   3D reconstruction

This section presents additional reconstruction results across various environments. As shown in Figure 8, 9, 10, 11, 12, 13, SinCro shows consistent qualitative and qualitative results compared to the baselines, particularly outperforming in the single-view inference.

In the multi-view input case, SinCro clearly captures small objects, such as the window handle in the window environment, the green cube in the push environment, or the hammerhead in the hammer environment, whereas all baselines struggle to render these parts. Despite using only three viewpoints (one primary input and two references), SinCro achieves competitive quantitative results compared to the baselines, which are provided with six viewpoints. Although the baselines show comparable PSNR, SSIM, and LPIPS to SinCro in the multi-view input case, this is essentially attributed to the baselines prioritizing the accurate reconstruction of non-salient parts of the image, which are not crucial for downstream tasks, and their RL performances are inferior to SinCro since capturing tiny objects in the scene is crucial for downstream RL tasks. Since our method is designed to take a handful of reference images rather than relying on multi-view images from densely located cameras, it requires significantly fewer viewpoints compared to the baselines. Therefore, the efficiency becomes even more pronounced with access to datasets from more diverse viewpoints, making the proposed method highly advantageous for memory efficiency and practical application in robotics.

In the single-view input case, SinCro outperforms the baselines both qualitatively and quantitatively. Moreover, our approach demonstrates consistent qualitative and quantitative results similar to those in the multi-view input case. In contrast, baselines experience significant rendering quality degradations in most environments, showing incomplete reconstruction, severe noise, jittering, teleportation, and confusion in distinguishing each different scene, leading to rapid degradations in PSNR and SSIM, and increases in LPIPS. These results are illustrated in the our supplementary video. Notably, 3D-NSR, which leverages time contrastive learning between image features, experiences less quantitative degradation than SNeRL and NeRF-RL. This suggests that time contrastive loss plays a key role in promoting view-invariancy in scene representations, encouraging them to be less sensitive to the number of viewpoints.

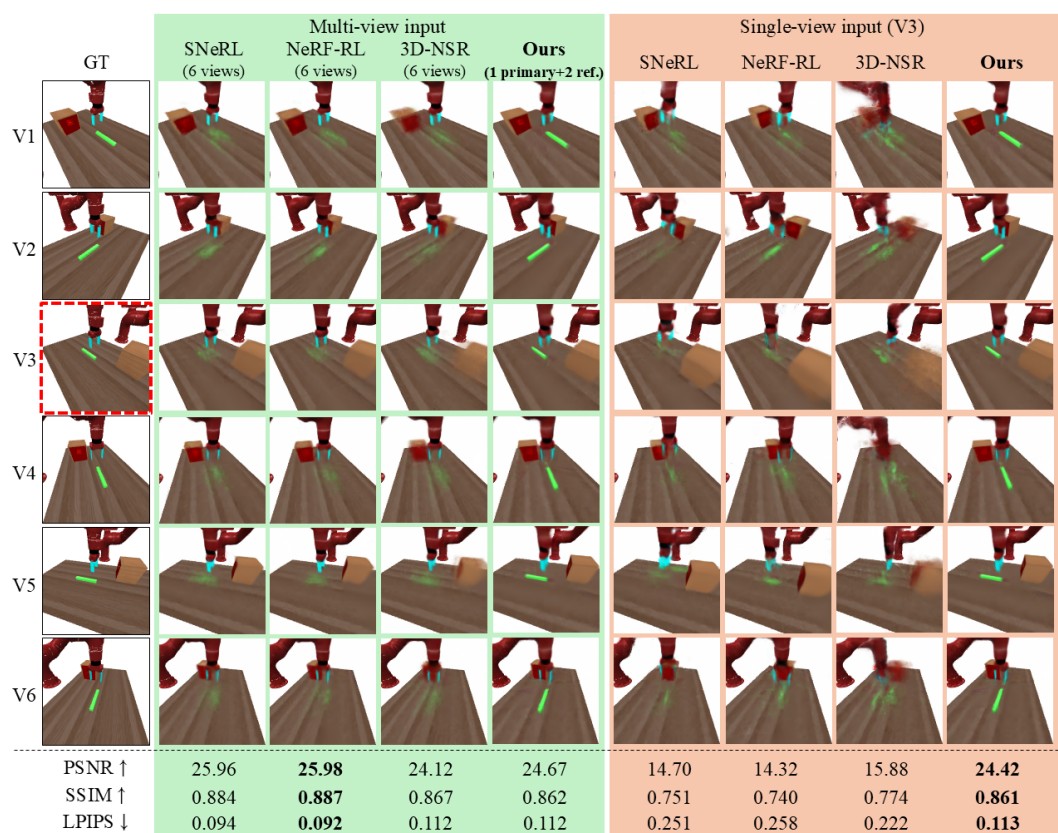

Figure 8: 3D volume rendering results in the peg-insert-side environment.

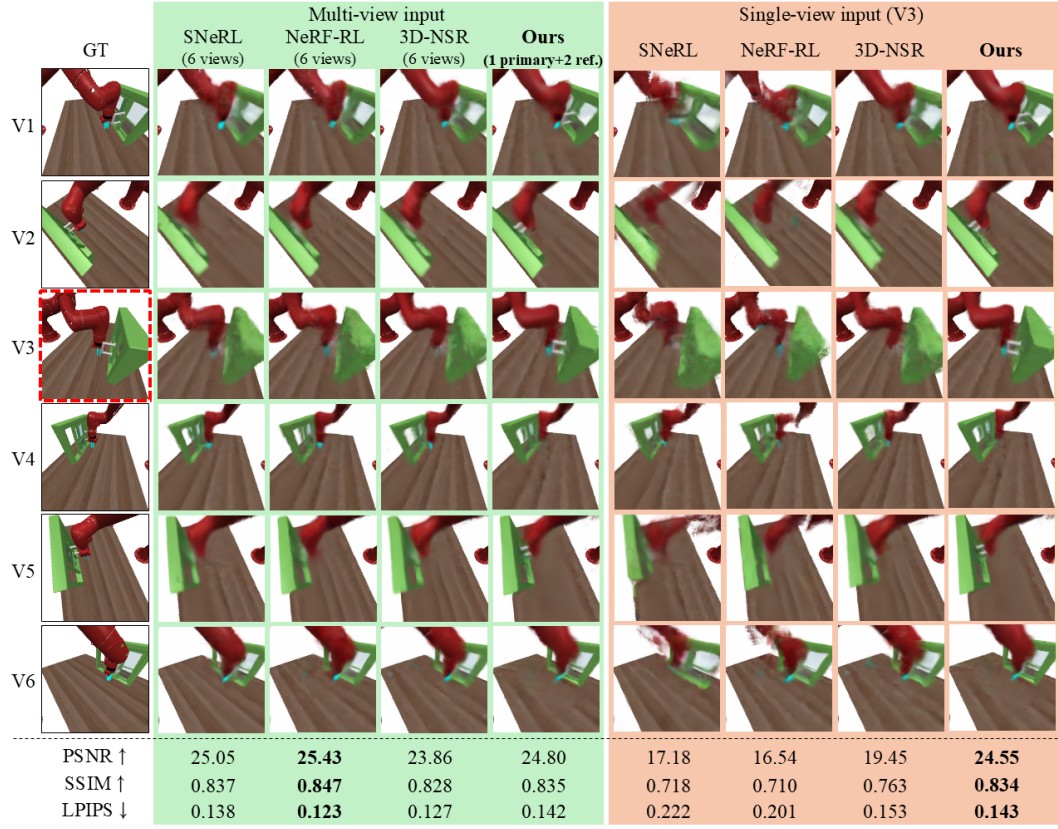

Figure 9: 3D volume rendering results in the window-open environment.

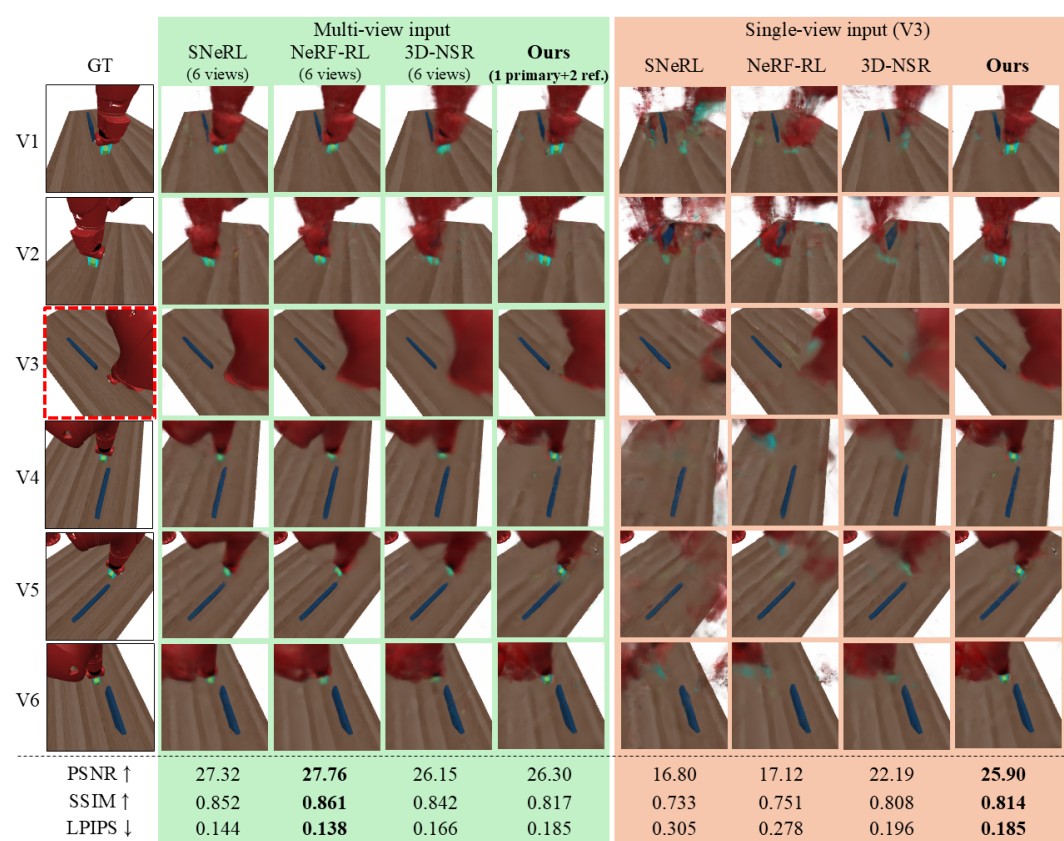

Figure 10: 3D volume rendering results in the push-wall environment.

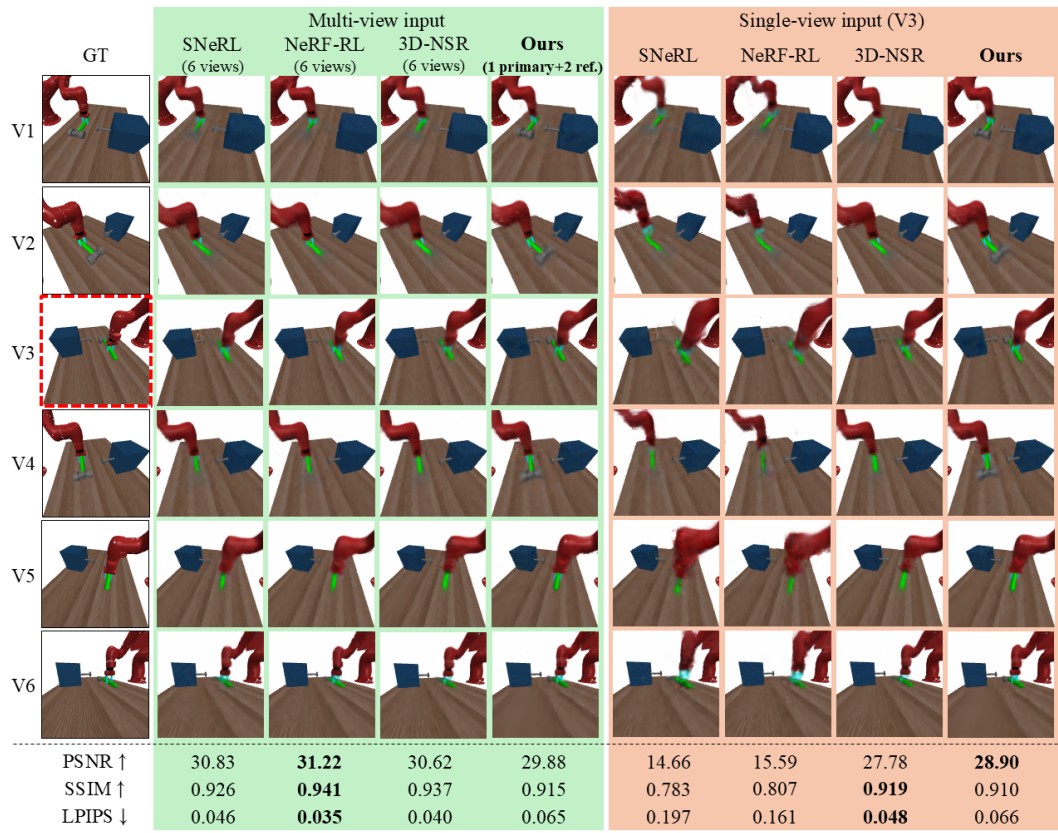

Figure 11: 3D volume rendering results in the hammer environment.

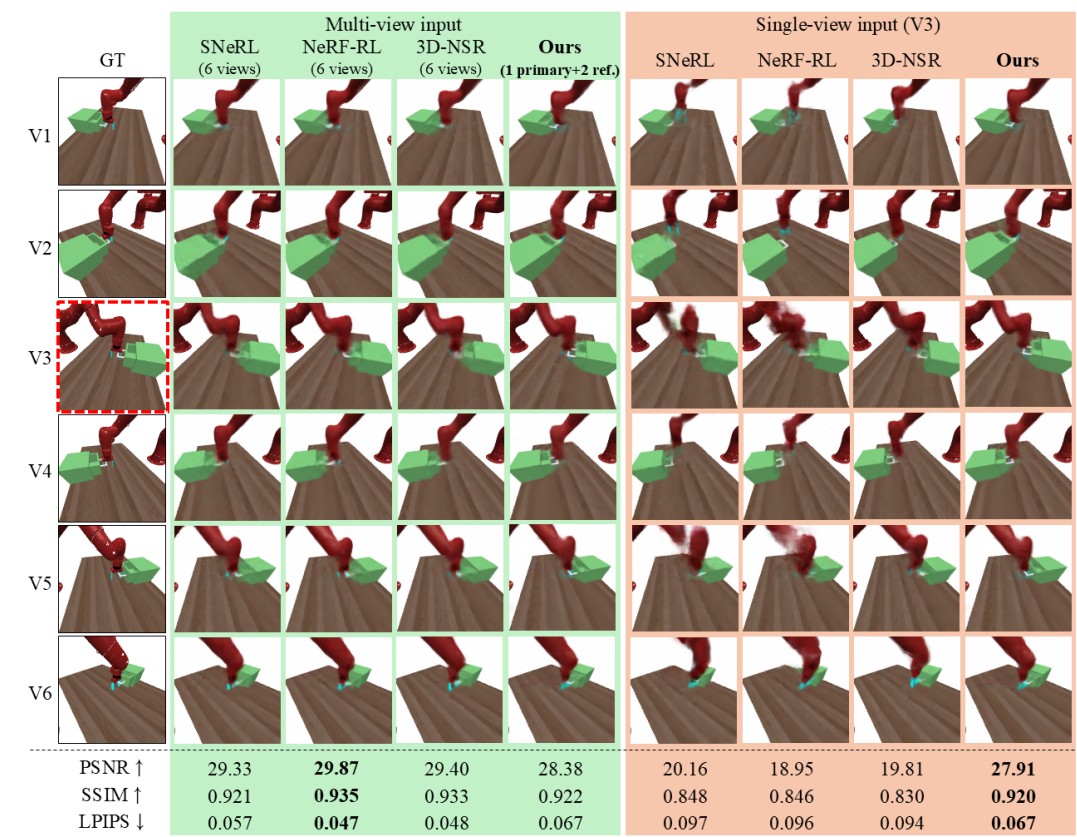

| | GT | Multi-view input | | | | Single-view input (V3) | | | |
|---|---|---|---|---|---|---|---|---|---|
| | | SNeRL (6 views) | NeRF-RL (6 views) | 3D-NSR (6 views) | **Ours** (1 primary+2 ref.) | SNeRL | NeRF-RL | 3D-NSR | **Ours** |
| PSNR ↑ | | 29.33 | **29.87** | 29.40 | 28.38 | 20.16 | 18.95 | 19.81 | **27.91** |
| SSIM ↑ | | 0.921 | **0.935** | 0.933 | 0.922 | 0.848 | 0.846 | 0.830 | **0.920** |
| LPIPS ↓ | | 0.057 | **0.047** | 0.048 | 0.067 | 0.097 | 0.096 | 0.094 | **0.067** |

Figure 12: 3D volume rendering results in the drawer-open environment.

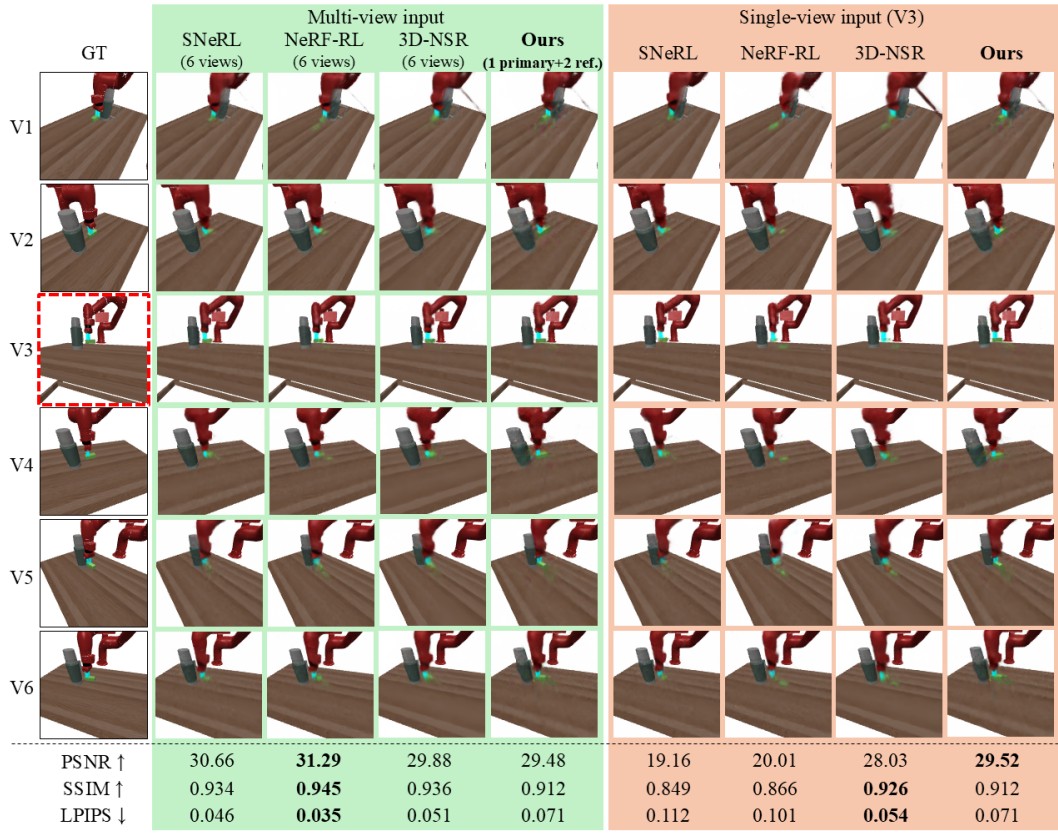

| | GT | Multi-view input | | | | Single-view input (V3) | | | |
|---|---|---|---|---|---|---|---|---|---|
| | | SNeRL (6 views) | NeRF-RL (6 views) | 3D-NSR (6 views) | **Ours** (1 primary+2 ref.) | SNeRL | NeRF-RL | 3D-NSR | **Ours** |
| PSNR ↑ | | 30.66 | **31.29** | 29.88 | 29.48 | 19.16 | 20.01 | 28.03 | **29.52** |
| SSIM ↑ | | 0.934 | **0.945** | 0.936 | 0.912 | 0.849 | 0.866 | **0.926** | 0.912 |
| LPIPS ↓ | | 0.046 | **0.035** | 0.051 | 0.071 | 0.112 | 0.101 | **0.054** | 0.071 |

Figure 13: 3D volume rendering results in the stick-push environment.

## B.2 THE EFFECT OF OBJECT-FOCUSED RAY SAMPLING FOR BASELINES

We apply the proposed object-focused ray sampling strategy to the baselines (SNeRL/NeRF-RL/3D-NSR w/ obj. sampling) to examine its effect. We perform image rendering using single-view input, and the results are shown in Figure 14. The baselines with our ray sampling strategy produce less blurry renderings of small objects, such as a green stick in the peg-insert-side environment or a small green cube in the push-wall environment, compared to the ones without the ray sampling strategy (SNeRL/NeRF-RL/3D-NSR original). However, these small objects' locations remain inaccurate, and the overall reconstruction results still exhibit severe noise, jittering of objects and the robot arm, or teleportation phenomena throughout the episode. This suggests that, even though our ray sampling strategy aids in reconstructing small but essential components of the environment, this strategy alone is insufficient to enhance 3D scene representation. In addition to the proposed ray sampling strategy, our proposed 3D scene representation learning scheme is required to acquire well-formed single-view 3D scene representations.

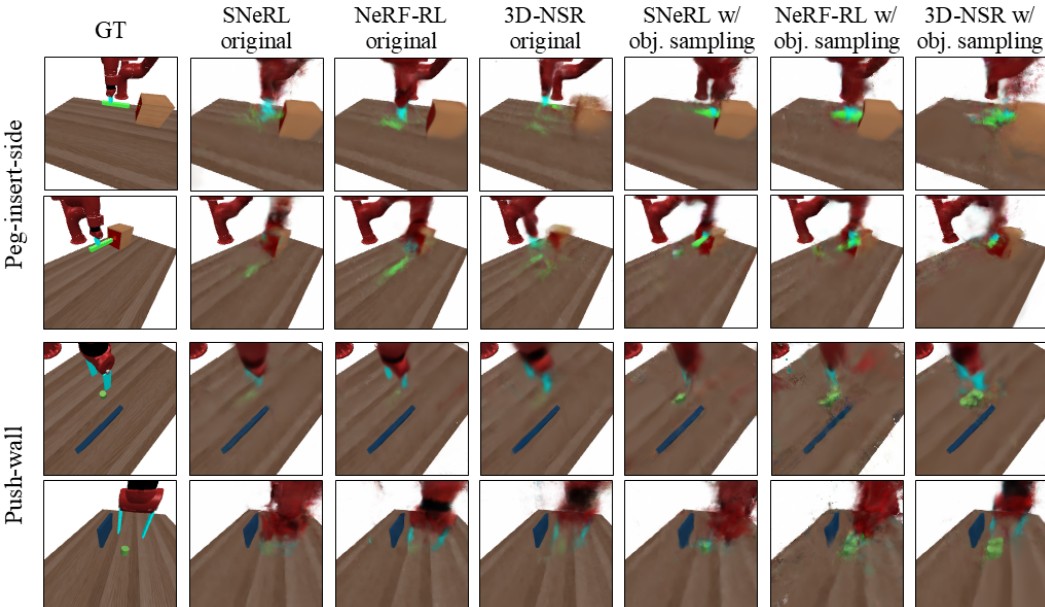

Figure 14: Qualitative results of the baselines with/without our proposed ray sampling strategy. The results are obtained using single-view input. Although the baselines with the ray sampling strategy achieve less blurry renderings of small objects, such as a green stick in the peg-insert-side task or a small green cube in the push-wall task, they still struggle to render the entire scenes, exhibiting inaccurate object and robot arm locations, as well as severe noise and jittering throughout the episode.

### B.3 ROBUSTNESS TO VIEWPOINT CHANGE

**Computing optimal transport matrix.**  A comparison of the optimal transport (OT) matrix between SinCro and baselines is presented in Figure 15. To compute the OT matrix, we utilize two observations of a task-solving trajectory from a default camera setting used in encoder pre-training and perturbed camera viewpoints. The perturbed camera continuously rotates around the default camera pose with maximum perturbations of five degrees in both azimuth and elevation. We collect all scene representations throughout the episode from each observation, then compare the two distributions using the equations provided in (Haldar et al., 2023; Hu et al., 2023). Due to the definition of the optimal transport, if the two distributions are similar, a scene representation $z_t'$ of the observation from the perturbed viewpoints will be transported to a representation $z_t$ of the observation with the same timestep $t$ obtained from the default camera pose. That is, bright colors (high values) will be aligned on the diagonal of the OT matrix, indicating strong correspondences between the two trajectories. Therefore, we can implicitly validate that the proposed method is robust to viewpoint perturbations as the scene representations are still aligned on the diagonal despite the perturbations. Compared to the baselines, SinCro demonstrates clear alignments of scene representations between the two observations. It indicates that the proposed method possesses 3D geometry-aware representations, enabling the agent to be robust in camera perturbations.

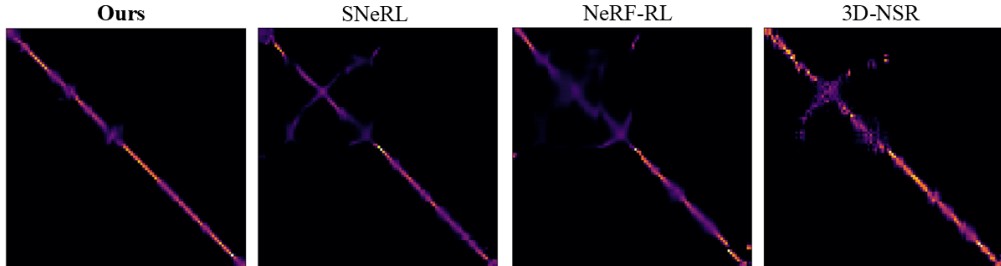

Figure 15: Comparison of the OT matrix in the stick-push environment (best viewed in the digital version).

**RL experiments for evaluating the robustness to viewpoint changes.**  We conducted view-point robust control experiments across all environments as outlined in the main script. As shown in Figure 16, the analysis in most environments aligns consistently with the results presented in the main text.

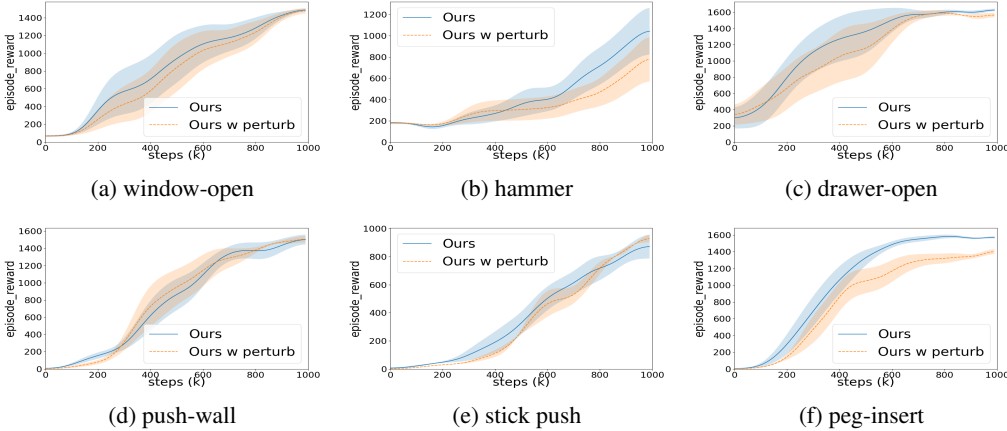

Figure 16: RL experiments for evaluating the robustness to viewpoint change.

### B.4 ABLATION STUDY

**3D volume rendering for ablation study.** Additional visual ablation results with single-view input are presented in Figure 17, 18. The results show the importance of each component in the proposed scene representation learning scheme. The reconstruction results of both models, trained without cross-view completion (**w/o cross-view**) and time contrastive loss (**w/o contrastive**), collapse into blurred images throughout the entire scenes. It indicates that cross-view completion and time contrastive loss are essential components of the proposed architecture for understanding 3D scene geometry and task progression. The ablation model without multi-view reconstruction (**w/o multi recon.**) occasionally shows errors in rendering object locations. For example, in Figure 17, the small green cube is inaccurately positioned in the V2 perspective when V3 is provided as a primary input, supporting that multi-view reconstruction enhances spatial understanding. Finally, without the object-focused ray sampling strategy (**w/o obj. sampling**), the model struggles to capture fine-grained details of small objects in the scene, such as the green cube in the push-wall environment and the green stick in the stick-push environment, both of which are crucial for success in down-stream RL tasks. Our ray sampling strategy significantly improves the render quality of tiny objects in the scene.

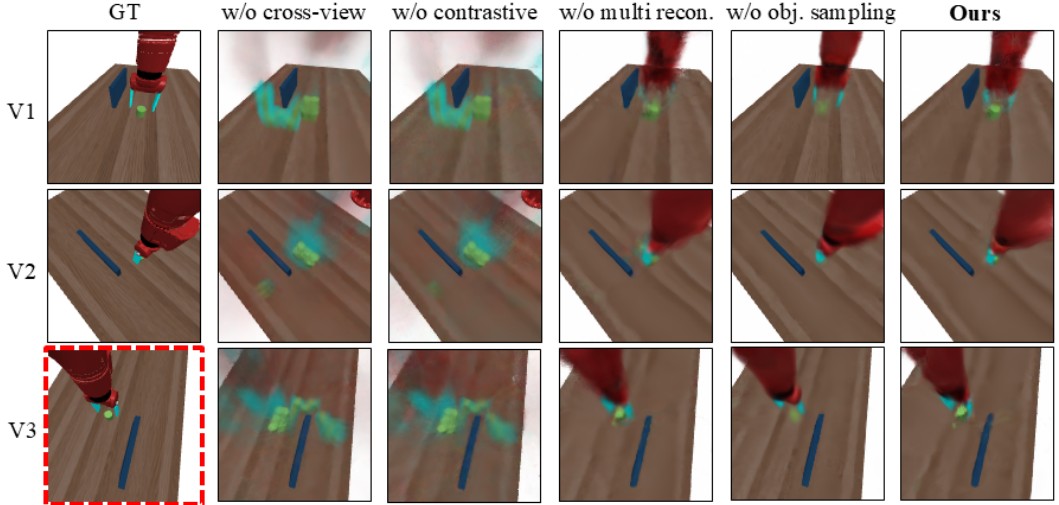

Figure 17: Visual ablation results with single-view input (outlined in V3) in push-wall environment.

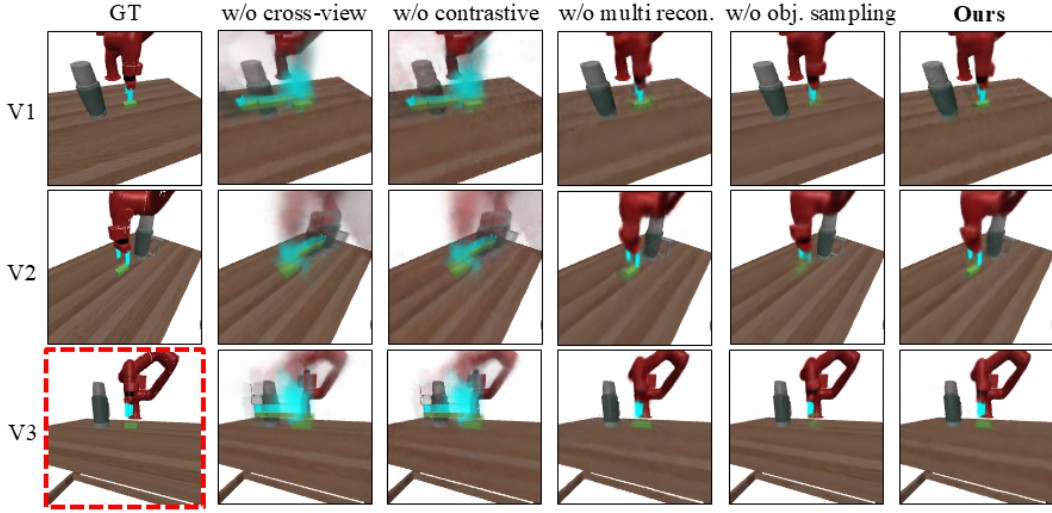

Figure 18: Visual ablation results with single-view input (outlined in V3) in stick-push environment.

**RL experiments for ablation study.** We conducted ablation studies across all environments as outlined in the main script. As shown in Figure 19, the analysis in most environments aligns consistently with the results presented in the main text.

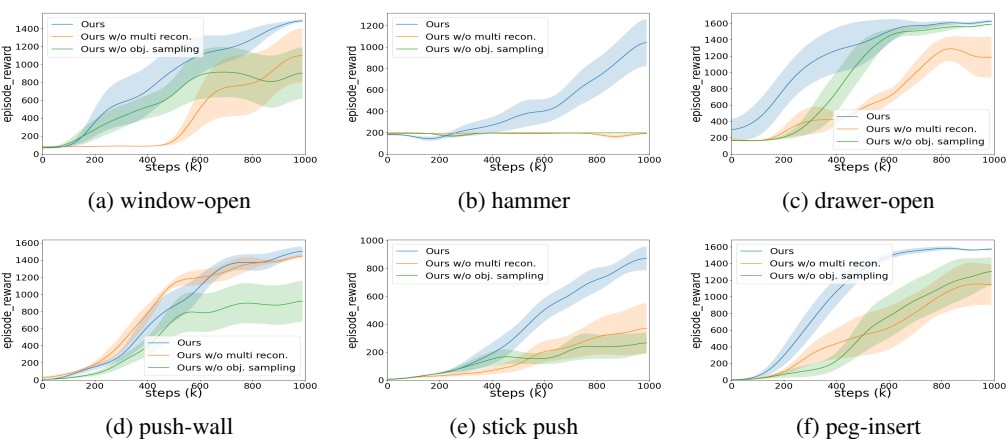

(a) window-open            (b) hammer            (c) drawer-open

(d) push-wall            (e) stick push            (f) peg-insert

Figure 19: RL experiments for ablation study

