# OpenReview forum: "Single-View 3D Representations for Reinforcement Learning by Cross-View Neural Radiance Fields"
_ICLR.cc/2025/Conference — Submitted to ICLR 2025_

### Official Review · Reviewer_JDLs · 2024-11-04

**Soundness:** 3
**Presentation:** 4
**Contribution:** 3
**Rating:** 6
**Confidence:** 4

**Summary:**

A 3D-aware representation learning approach is presented in which posed multiview data is leveraged to learn view-invariant representations from images. These representations can be used as auxiliary input to an RL policy, where it is shown they achieve superior performance relative to other such baselines.

**Strengths:**

Learning viewpoint invariant embeddings is an important problem in robotics as prior work has shown the sensitivity of robot policies to out-of-domain camera viewpoints.

The experimental evaluation and ablation study as well as qualitative analysis are quite thorough and nice to see.

**Weaknesses:**

* The proposed representation requires synchronized multiview video data to be trained, so it can only be trained on limited data. It would be good to compare against embeddings such as DinoV2 which do not have explicit geometry-aware nature but can be trained on a lot more data and probably have a notion of “view-invariance” to some degree due to their training strategy.

* Table 1 is a bit misleading. While during deployment the proposed algorithm can indeed be run on single view input, if I’m not mistaken during training of the actual embedding the requirement is still for posed multiview data. Perhaps it would be better to disentangle the deployment and training stages in this Table for the proposed method and for baselines as applicable.

* I think the paper focuses slightly too much on the few- or single-view reconstruction results visually and wr.t. view synthesis metrics, which I don’t think is particularly informative. Single- and few-view reconstruction is a huge field by itself and there are much stronger baselines to compare against if this is the goal such as PixelNeRF, NeRDi, GS-LRM, ZeroNVS, Cat3D, Reconfusion, etc. etc. The goal of the paper is not to solve single or few view 3D reconstruction but to learn 3D-aware representations for downstream RL.

**Questions:**

Of course, it’s not necessary to compare, but it may be good to discuss some concurrent related works, such as Dreamitate, VISTA, RoVi-AUG, which leverage generative models to learn view-invariant RL policies.

---

> ### Author Response · Authors · 2024-11-19
> **Author Response 1**
>
> **Comment 1:**
>
> The proposed representation requires synchronized multiview video data to be trained, so it can only be trained on limited data. It would be good to compare against embeddings such as DinoV2 which do not have explicit geometry-aware nature but can be trained on a lot more data and probably have a notion of “view-invariance” to some degree due to their training strategy.
>
> **Response 1:**
>
> Thank you for the suggestion. We conducted experiments using DINOv2 embeddings as the representation for our RL setup. However, the results were not favorable.
>
> Anonymous link to RL with DINO-v2:
>
> https://drive.google.com/drive/folders/13WU3Rq2vZvt68vlgjdVWngLL52skPuM_?usp=sharing
>
> We believe there are two primary reasons for this outcome. First, DINOv2 is trained on real-world data, which introduces a significant sim-to-real gap in our experimental setup. This mismatch in data domains likely hampers performance in the simulated environment we use for RL.
>
> Second, while DINO has been applied successfully in contexts such as temporal segmentation of robot trajectory for imitation-based approach [1], the requirements for RL differ significantly from methods like imitation learning. In RL, particularly with Bellman operator-based optimization, simply incorporating an off-the-shelf pre-trained backbone does not guarantee effective learning. Instead, RL demands a careful integration of embeddings with the underlying optimization process. Related works, such as [2] highlight that the effectiveness of a pre-trained vision model highly depends on the downstream policy learning method, and the pre-trained representations often require specific adaptations to align with RL's optimization dynamics.
>
> This observation reinforces the need for representation learning approaches specifically designed for RL, as demonstrated in our method.
>
> [1] Wan, Weikang, et al. "Lotus: Continual imitation learning for robot manipulation through unsupervised skill discovery." *2024 IEEE International Conference on Robotics and Automation (ICRA)*. IEEE, 2024.
>
> [2] Hu, Yingdong, et al. "For pre-trained vision models in motor control, not all policy learning methods are created equal." *International Conference on Machine Learning*. PMLR, 2023.
>
> **Comment 2:**
>
> Table 1 is a bit misleading. While during deployment the proposed algorithm can indeed be run on single view input, if I’m not mistaken during training of the actual embedding the requirement is still for posed multiview data. Perhaps it would be better to disentangle the deployment and training stages in this Table for the proposed method and for baselines as applicable.
>
> **Response 2:**
>
> Thank you for your suggestion. We have modified the table to separate the pre-training and deployment phases for a clearer comparison.
>
> Anonymous link for modified table
>
> https://drive.google.com/file/d/1fhHpJtBrUcDoLxycD13AnSD-aVv6RKX6/view?usp=sharing
>
> The revised manuscript has been uploaded.
>
> **Comment 3:**
>
> I think the paper focuses slightly too much on the few- or single-view reconstruction results visually and wr.t. view synthesis metrics, which I don’t think is particularly informative. Single- and few-view reconstruction is a huge field by itself and there are much stronger baselines to compare against if this is the goal such as PixelNeRF, NeRDi, GS-LRM, ZeroNVS, Cat3D, Reconfusion, etc. etc. The goal of the paper is not to solve single or few view 3D reconstruction but to learn 3D-aware representations for downstream RL.
>
> **Response 3:**
>
> Thank you for your insightful feedback. We acknowledge that the current manuscript seems slightly focused on the single-view reconstruction results, which we think are indirect validations for our learned latent representation $z_t$’s 3D understanding. We have uploaded a revised manuscript to make clear that our work’s focus is learning effective representation for RL.

---

> > ### Author Response · Authors · 2024-11-19
> > **Author Response 2**
> >
> > **Question 1:**
> >
> > Of course, it’s not necessary to compare, but it may be good to discuss some concurrent related works, such as Dreamitate, VISTA, RoVi-AUG, which leverage generative models to learn view-invariant RL policies.
> >
> > **Answer 1:**
> >
> > Thank you for your comment. There is a similar question from another reviewer 4UuN, so refer to Comment 1 for this reviewer if you are interested.
> >
> > Just so you know, both RoVi-Aug and VISTA use ZeroNVS for data augmentation by synthesizing novel-view images, focusing on generating novel-view data rather than learning effective representations. Consequently, we believe these methods can serve as **complementary** components, as we could replace the multi-view dataset assumption with a dataset augmented by ZeroNVS, as in RoVi-Aug and VISTA, to further enhance our approach.
> >
> > To provide preliminary insights, we conducted data augmentation experiments to evaluate the policy’s robustness to viewpoint changes, similar to RoVi-Aug and VISTA. As mentioned in Response 3 for reviewer 4UuN, ZeroNVS does not produce reasonable zero-shot synthesis results. Therefore, we used ground truth images obtained from simulation as a proxy for the data augmentation effect of ZeroNVS (i.e., assuming ZeroNVS synthesizes 100% accurate images). Specifically, we performed an RL experiment similar to Section 5.3, while following CNN+view randomization described in the manuscript, but with 30 viewpoints. The results are available in the following anonymous link:
> >
> > https://drive.google.com/drive/folders/1oUNOOcUNQGkIcCjlpFKT_B5tyhr-HJ3J?usp=sharing
> >
> > As shown in these figures, our method outperforms this baseline despite utilizing images from significantly fewer viewpoints (Ours: 6 views, Baseline: 30 views). We attribute this result to the proposed 3D-aware representation learning scheme, which enhances the encoder’s implicit understanding of the 3D world. This finding highlights that simply increasing the number of viewpoints is not always the optimal approach; instead, carefully designed representation learning schemes play a far more critical role.
> >
> > If you have any questions or need more discussion, please let us know. We would be happy to improve our work based on your valuable feedback.

---

> > > ### Author Response · Authors · 2024-11-24
> > > **Remind**
> > >
> > > We sincerely thank all the reviewers for reviewing our work and providing constructive feedback. We hope that our response has adequately addressed your comments. If you have any remaining questions (existing or new ones) that we can address in our follow-up response to improve your opinion about our work, please do not hesitate to provide additional feedback in the comments. It would be greatly appreciated if we could have more discussions about our work which would provide valuable insights towards further developing our research into a meaningful contribution in the RL domain.

---

### Official Review · Reviewer_M7W6 · 2024-11-04

**Soundness:** 3
**Presentation:** 2
**Contribution:** 2
**Rating:** 5
**Confidence:** 2

**Summary:**

This paper presents SinCro, a framework for learning 3D-aware representations for reinforcement learning that can operate with single-view inputs during deployment. The key innovation is combining a masked ViT encoder with a latent-conditioned NeRF decoder, trained through cross-view completion and time contrastive learning. The method enables single-view 3D representation inference without requiring camera calibration during deployment, while previous approaches typically needed multi-view inputs or calibrated cameras.

**Strengths:**

- The technical approach is well-motivated and addresses a practical limitation of existing 3D representation learning methods for RL - the requirement for multi-view or calibrated cameras during deployment
- The empirical results demonstrate the method works as intended, achieving comparable performance to multi-view baselines while requiring only single-view input

**Weaknesses:**

My primary concerns are:

- The evaluation is limited to MetaWorld environments, which are relatively simple by 2024 standards. Testing on more complex manipulation scenarios would strengthen the paper. There are a lot of other simulated environments like RLBench. Can you explain why MetaWorld is used?
- The quantitative results in Figure 3 show an apparent contradiction - NeRF-RL achieves higher PSNR despite producing visibly blurrier reconstructions. This needs better explanation. Can you explain why NeRF-RL images are blurry but the PSNR is higher than SinCro?

Some additional comments

- The figures could be improved - Figure 2 is a PNG instead of vector graphics which reduces quality

**Questions:**

See weaknesses.

---

> ### Author Response · Authors · 2024-11-19
> **Author Response 1**
>
> **Comment 1:**
>
> The evaluation is limited to MetaWorld environments, which are relatively simple by 2024 standards. Testing on more complex manipulation scenarios would strengthen the paper. There are a lot of other simulated environments like RLBench. Can you explain why MetaWorld is used?
>
> **Response 1:**
>
> Thank you for your comments. Our primary focus in this work is on the algorithmic development of 3D representation learning for RL, as this is the first attempt at a single-view inference framework for 3D-aware representation, to the best of our knowledge. So, we followed the environment setup from prior work [1], while modifying the environment to a more realistic and challenging setup, incorporating additional textures from elements like a table and robot body. However, it is not an inherent limitation of our work, and we can consider performing experiments in other simulated environments like RLBench.
>
> [1] Shim, Dongseok, Seungjae Lee, and H. Jin Kim. "Snerl: Semantic-aware neural radiance fields for reinforcement learning." *International Conference on Machine Learning*. PMLR, 2023.
>
> **Comment 2:**
>
> The quantitative results in Figure 3 show an apparent contradiction - NeRF-RL achieves higher PSNR despite producing visibly blurrier reconstructions. This needs a better explanation. Can you explain why NeRF-RL images are blurry but the PSNR is higher than SinCro?
>
> **Response 2:**
>
> As you rightly pointed out, NeRF-RL achieves a higher PSNR in the multi-view input setting. This is because slight blurring at edges or differences in high-frequency details can reduce PSNR without significantly impacting perceived image quality. For example, NeRF-RL renders sharp boundaries in non-salient areas, such as table textures (which occupy most of the pixels) and the edges of the robot arm, whereas our method shows minor degradation in these areas. Also, blurred regions, such as the green peg in Figure 3, constitute only a small portion of a scene and thus have a limited impact on the overall PSNR while significantly contributing to the downstream RL performance.
>
> **Comment 3:**
>
> The figures could be improved - Figure 2 is a PNG instead of vector graphics which reduces quality.
>
> **Response 3:**
>
> Thank you for your feedback. We replaced Figure 2 with a vector graphic format to ensure higher quality and better readability.
>
> If you have any questions or need more discussion, please let us know. We would be happy to improve our work based on your valuable feedback.

---

> > ### Author Response · Authors · 2024-11-24
> > **Remind**
> >
> > We sincerely thank all the reviewers for reviewing our work and providing constructive feedback. We hope that our response has adequately addressed your comments. If you have any remaining questions (existing or new ones) that we can address in our follow-up response to improve your opinion about our work, please do not hesitate to provide additional feedback in the comments. It would be greatly appreciated if we could have more discussions about our work which would provide valuable insights towards further developing our research into a meaningful contribution in the RL domain.

---

> > > ### Comment · Reviewer_M7W6 · 2024-11-26
> > >
> > > Thanks for the response. I don't have further questions.

---

> > > > ### Author Response · Authors · 2024-11-29
> > > >
> > > > We sincerely appreciate all your efforts during the review process. We believe we have addressed all the comments and hope the updated manuscript better reflects the quality and impact of our work. If our response is not sufficiently clear to reconsider the score, please do not hesitate to let us know. We would be happy to discuss further to improve our work.

---

### Official Review · Reviewer_jiZx · 2024-11-06

**Soundness:** 2
**Presentation:** 2
**Contribution:** 2
**Rating:** 3
**Confidence:** 4

**Summary:**

Summary
This paper introduces a 3D representation reinforcement learning (RL) framework that utilizes a single view for inference. The downstream RL process leverages the latent code derived from the single image as input for the RL tasks.

**Strengths:**

Strengths

The results of the proposed method demonstrate superior performance compared to previous relative methods.

**Weaknesses:**

Weaknesses

The writing quality requires improvement, for example in line 300, where the meaning is unclear. The preceding sentence discusses a reinforcement learning (RL) algorithm, but the subsequent sentence shifts focus to data shuffling, creating a disjointed narrative. Additionally, this sentence is ambiguous  and difficult to comprehend (eg, why `randomize viewpoint` but not `random pick a viewpoint` ).

In line 93, the authors do not clarify the concept of a calibrated camera, both in this section and in subsequent ones. Additionally, the process for computing the (x,d) values during rendering is not explained. Therefore, the claim that the proposed method operates 'without requiring camera calibration' is misleading; instead, it could be interpreted that 'camera calibration is addressed through overfitting.' It appears that the authors are utilizing an absolute camera pose along with a fixed intrinsic matrix. Consequently, the image encoder and neural radiance fields (NeRF) are effectively learning a fixed RGB->pose mapping. This principle is referenced in [1] and may lead to poor generalization. Furthermore, recent multi-view stereo (MVS) reconstruction models demonstrate that a calibration matrix is not essential for creating MVS 3D models. The authors should explore relevant literature in the domains of lightweight regression models (LRM), single-view LRMs, LRM with Gaussian distributions, and indoor LRM-like methodologies.

The image encoder and NeRF appear to be overfitting to the given dataset, similar to previous dynamic NeRF approaches that attempt to learn a mapping of f(x,d,t)=c,\rho. The latent variable z in the proposed method effectively serves as a latent code encompassing (t, action, state, pose, intrinsic parameters, and object). For instance, in Figure 3, if the proposed method utilizes only view V3 as input, it can accurately recover the clearly marked red annotation on the box, which is not visible in view V3. To the best of my knowledge, no existing methods—whether single view to 3D, MVS to 3D, learning-based, diffusion-based, for objects, indoors, or outdoors— can achieve it whtiout overfitting.

Several baseline comparisons are missing. Since the proposed method aims to illustrate the effectiveness of a single-view latent 3D representation for RL processes, it is essential for the authors to include baselines that utilize explicit 3D representations, such as depth maps or 3D volumes, as presented in the recent conference proceedings.

[1] PoseNet: A Convolutional Network for Real-Time 6-DOF Camera Relocalization

**Questions:**

1. The author should explain more details about the camera calibration.
2. The author should add some baselines with explicit 3d reapresentation in RL.
3. The visualization result of snerl looks much worse than it original paper, it will be great to see the visualization on the same env and setting.

---

> ### Author Response · Authors · 2024-11-19
> **Author Response 1**
>
> **Comment 1:**
>
> The writing quality requires improvement...
>
> **Response 1:**
>
> We acknowledge that there are some unclear sentences as you rightly pointed out. For example, ‘randomly select a viewpoint’ is more clear expression compared to ‘randomizing viewpoints’. We have uploaded a new manuscript with the revised parts highlighted in magenta color.
>
> **Comment 2:**
>
> In line 93, the authors do not clarify the concept of a calibrated camera, ...., Consequently, the image encoder and neural radiance fields (NeRF) are effectively learning a fixed RGB->pose mapping.
>
> **Response 2:**
>
> First, we would like to kindly ask the reviewer to refer to the Common Response that addresses some misunderstandings regarding our work. We would like to note that our work is **NOT** a high-quality novel view synthesis framework without calibration, and the 3D reconstruction results in the manuscript are included as indirect validations for our learned latent representation $z_t$’s 3D understanding. In this work, we have focused on learning effective latent representation for downstream image-based RL, which itself is a huge field in RL.
>
> Our work consists of 2 steps. 1) pre-train the 3D scene encoder to extract effective representation by leveraging NeRF (require calibrated camera at this phase for volume rendering), 2) perform inference on the pre-trained encoder and utilize the output representation of the encoder as an input for downstream RL. At this phase, we do **NOT** perform rendering (In Figure 1, the deployment phase does not include NeRF). For this reason, we said that camera calibration is not required **during the downstream RL phase,** while other prior representation learning works in RL require synchronized multi-view input or camera pose even during the downstream RL phase (Table 1).
>
> We acknowledge that the manuscript may have been slightly unclear on the above points, and we have uploaded a revised manuscript to clarify them. Regarding overfitting, please refer to the Response 4.
>
> **Comment 3:**
>
> Furthermore, recent multi-view stereo (MVS) reconstruction models demonstrate that a calibration matrix is not essential for creating MVS 3D models. The authors should explore relevant literature in the domains of lightweight regression models (LRM), single-view LRMs, LRM with Gaussian distributions, and indoor LRM-like methodologies.
>
> **Response 3:**
>
> We assume your comment refers to a Large Reconstruction Model [1], not a Lightweight Regression Model, since most keywords you mentioned are addressed in [1]-related works. If we misunderstood your comments, please kindly let us know.
>
> Similar to Response 2, we would like to emphasize that our work is not a high-quality novel view synthesis framework without calibration. Our focus is on learning effective latent representations for downstream image-based RL. Since the referenced works mostly focus on 3D reconstruction generalization capability or novel-view synthesis using single-view frameworks, we believe they fall outside the scope of our work and are not direct baselines for comparison.
>
> Nevertheless, we investigated the rendering results of LRM. The novel-view synthesis results of LRM were significantly distorted when perturbations from the input viewpoint exceeded approximately 10 degrees, as illustrated in the following anonymous link:
>
> https://drive.google.com/file/d/1AXY0UjwR2ctwuqNSxSPskNuaL2hWNsnr/view?usp=sharing
>
> [1] Hong, Yicong, et al. "Lrm: Large reconstruction model for single image to 3d." *arXiv preprint arXiv:2311.04400* (2023).

---

> > ### Author Response · Authors · 2024-11-19
> > **Author Response 2**
> >
> > **Comment 4:**
> >
> > The image encoder and NeRF appear to be overfitting to the given dataset. ... To the best of my knowledge, no existing methods can achieve it without overfitting.
> >
> > **Response 4:**
> >
> > As you rightly pointed out, the NeRF model in our work exhibits some degree of overfitting at the texture level. For example,  in an environment like the peg shown in Figure 3, the NeRF model might have memorized that the hole’s color and texture are red and square. However, for RL-based robotics applications, the primary concern is the **3D spatial information rather than the visual appearance itself**. That is, understanding where the components (e.g. box, peg) of the scene are located in 3D space is far more crucial for solving RL-based robotic tasks. In this aspect, we think slight texture-level overfitting is allowable as long as the model can recognize the 3D spatial information. This principle is widely adopted and referenced in prior representation learning works for RL [1,2].
> >
> > Since there are multiple episodes with randomly initialized object positions in each episode, if the proposed model was fully overfitted in terms of spatial awareness, the object positions would be fixed in the renderings from other viewpoints. However, our model identifies varying object locations in each episode, based on the input image. This indicates that the model has an implicit understanding of where the objects are located in 3D space based solely on the given single-view image input. In other words, the model generalizes the spatial positioning of objects while overfitting the textures representing those objects.
> >
> > [1] Seo, Younggyo, et al. "Multi-view masked world models for visual robotic manipulation." *International Conference on Machine Learning*. PMLR, 2023.
> >
> > [2] Shim, Dongseok, Seungjae Lee, and H. Jin Kim. "Snerl: Semantic-aware neural radiance fields for reinforcement learning." *International Conference on Machine Learning*. PMLR, 2023.
> >
> > **Comment 5:**
> >
> > Several baseline comparisons are missing. Since the proposed method aims to illustrate the effectiveness of a single-view latent 3D representation for RL processes, it is essential for the authors to include baselines that utilize explicit 3D representations, such as depth maps or 3D volumes, as presented in the recent conference proceedings.
> >
> > **Response 5:**
> >
> > Thank you for your comments. First, as noted in lines 124-127 of the revised manuscript, we would like to clarify that explicit 3D representation methods such as 3D volumes or point clouds require **additional requirements** such as synchronized and calibrated RGB-D cameras to get depth and point clouds **even during the downstream tasks**. These requirements introduce complexities that differ from our approach (RGB only, single-view). Therefore, we believe the direct comparison is less straightforward, and we have included relevant baselines that align with our setting in the current manuscript.
> >
> > Following the reviewer's suggestion, we conducted additional experiments to evaluate an explicit 3D representation-based baseline in our RL set-up. Specifically, we follow the same RL training process in our work, while replacing our proposed 3D scene encoder with the encoder architecture from an imitation learning approach [1] that processes point cloud inputs derived from depth data. For a fair comparison, we used only the point cloud and visual features, excluding language instructions and proprioceptive inputs for the encoder. Despite these efforts, the results were not favorable.
> >
> > Anonymous link to the RL results with the point cloud-based encoder:
> >
> > https://drive.google.com/drive/folders/13iosZFtnhyexUyQw89rulf03kcXHg7DW?usp=sharing
> >
> > We attribute this to the unique requirements of RL that differ significantly from tasks like imitation learning. In RL, particularly with Bellman operator-based optimization, simply incorporating an off-the-shelf encoder architecture of another paradigm, such as imitation learning, does not guarantee effective learning. This challenge has been highlighted in related works, such as [2], which emphasize the effectiveness of the backbone encoder highly depends on the downstream policy learning method, often requiring specific adaptations to align with RL's optimization dynamics.
> >
> > This observation reinforces the need for representation learning approaches specifically designed for RL, as demonstrated in our method.
> >
> > [1] Ke, Tsung-Wei, Nikolaos Gkanatsios, and Katerina Fragkiadaki. "3d diffuser actor: Policy diffusion with 3d scene representations." *arXiv preprint arXiv:2402.10885* (2024).
> >
> > [2] Hu, Yingdong, et al. "For pre-trained vision models in motor control, not all policy learning methods are created equal." *International Conference on Machine Learning*. PMLR, 2023.

---

> ### Author Response · Authors · 2024-11-19
> **Author Response 3**
>
> **Question 1:**
>
> The author should explain more details about the camera calibration.
>
> **Answer 1:**
>
> We addressed this question in Response 2. If it remains unclear, please let us know.
>
> **Question 2:**
>
> The author should add some baselines with explicit 3d representation in RL.
>
> **Answer 2:**
>
> We addressed this question in Response 5. If it remains unclear, please let us know.
>
> **Question 3:**
>
> The visualization result of snerl looks much worse than it original paper, it will be great to see the visualization on the same env and setting.
>
> **Answer 3:**
>
> As mentioned in the revised manuscript (lines 309-311), the experimental environment used in this work is a modified version of the SNeRL environment, designed to provide a more realistic and challenging setup by incorporating additional textures from elements such as a table and robot body.
>
> The reproduced results of SNeRL confirm that it can capture objects in the environment of the SNeRL paper, as illustrated in the anonymous link provided:
>
> https://drive.google.com/file/d/1ePRBSd16gDrBkSF737O0L8HSZ1LYn8t2/view?usp=sharing
>
> We believe that SNeRL performed well in the original paper since it was tested under simpler environments. However, it struggles as the complexity of the scene and the number of viewpoints increase. This limitation stems from its reliance on a simple CNN and the absence of masked reconstruction and cross-view completion strategies, making it unsuitable for scaling to more complex scenes and a larger number of viewpoints.
>
> If you have any questions or need more discussion, please let us know. We would be happy to improve our work based on your valuable feedback.

---

> > ### Author Response · Authors · 2024-11-24
> > **Remind**
> >
> > We sincerely thank all the reviewers for reviewing our work and providing constructive feedback. We hope that our response has adequately addressed your comments. If you have any remaining questions (existing or new ones) that we can address in our follow-up response to improve your opinion about our work, please do not hesitate to provide additional feedback in the comments. It would be greatly appreciated if we could have more discussions about our work which would provide valuable insights towards further developing our research into a meaningful contribution in the RL domain.

---

> ### Author Response · Authors · 2024-11-29
> **Additional response to the Comment 5**
>
> We have conducted additional experiments to compare our method with another explicit 3D representation baseline, GNFactor [3], which utilizes posed RGB-D to construct 3D volumes and reconstruct the images from multiple viewpoints. Due to the limited response window, we could not conduct downstream RL experiments with this explicit representation. Thus, we only perform the pre-training phase, which could be utilized as indirect validations for the learned representation’s 3D understanding.
>
> Anonymous link for the single-view volume rendering results: https://drive.google.com/file/d/1G3zERn-KFOwYT09ASIHK6Xvfp0AJRlQU/view?usp=sharing
>
> The results demonstrate that the explicit representation baseline also struggles to capture fine-grained object details, such as the green peg, despite using depth information. As shown in the RL experiments in the manuscript, capturing these details is correlated to the downstream RL performance. Therefore, we expect that our method has the potential to outperform the explicit 3D representation baseline in downstream RL tasks. Furthermore, since this explicit representation baseline was originally designed to work in a single fixed viewpoint, the RL performance would further degrade as the input viewpoint varies.
>
> [3] Yanjie Ze, et al., “Multi-Task Real Robot Learning with Generalizable Neural Feature Fields.” CoRL, 2023.

---

> > ### Comment · Reviewer_jiZx · 2024-12-01
> >
> > Thanks for the clarification. I don't have further questions.

---

### Official Review · Reviewer_amkh · 2024-11-09

**Soundness:** 3
**Presentation:** 2
**Contribution:** 2
**Rating:** 5
**Confidence:** 3

**Summary:**

This paper introduces a framework that generates 3D-aware representations from single-view camera inputs, which can be rendered into observations for training RL models. The 3D reconstruction model uses an autoencoder architecture, with a masked ViT as the encoder and a latent-conditioned NeRF as the decoder, trained with cross-view completion objectives. Experimental results demonstrate that the proposed method greatly improves the RL agent's performance for complex tasks.

**Strengths:**

* The proposed method can reconstruct 3D scene representation from single-view images, eliminating the need for multi-sensor setup and calibration for learning downstream RL algorithms.
* By using an autoencoder architecture to learn the NeRF representation, it bypasses the time-consuming optimization required in classical NeRF reconstruction methods and potentially predicts occluded regions, unlike traditional NeRF approaches.
* The authors conduct extensive experiments to demonstrate that the proposed methods achieve superior performance for both volume rendering and downstream RL algorithms such as DrM.

**Weaknesses:**

* The time contrastive loss (Eqn. 3) repulses state features at different timesteps. However, this does not hold for static scenes where the actor remains stationary between timesteps $t$ and timestep $t^\prime$.
* The 3D encoder-decoder model $\Omega_\theta$ is trained on multi-view images, with scene representation $z_t = \Omega_\theta(O_{t-2:t}^i, O_{t-2:t}^{r_1}, \cdots, O_{t-2:t}^{r_K})$. How can it generalize when the inputs are from the same viewpoint, as in $z_t = \Omega_\theta(O_{t-2:t}^i, [O_{t-2:t}^i,] * K)$ (line 291)?
* Table 1 claims that the proposed method does not require camera calibration. However, camera poses are needed to render multi-review reconstruction from $z_t$, making this claim inaccurate.
* In the volume rendering experiments, the authors should also include comparisons with NeRF baselines for sparse views, such as RegNeRF, pixelNeRF, etc.
* The RL experiments are conducted on toy environments. It would be valuable to see the method's performance in real-world robotic settings.

**Questions:**

See the weakness above.

---

> ### Author Response · Authors · 2024-11-19
> **Author Response 1**
>
> **Comment 1:**
>
> The time contrastive loss (Eqn. 3) repulses state features at different timesteps. However, this does not hold for static scenes where the actor remains stationary between timesteps $t$ and timestep $t’$.
>
> **Response 1:**
>
> Thank you for your insightful comments. As you mentioned correctly, repulsing state features at different timesteps may not hold for totally static episodes. However, since the RL algorithm itself is inherently designed to explore the environment to maximize the reward, the agent continues to move during the episode rollout, preventing static episodes. Furthermore, this time contrastive loss has been widely adopted and validated in other robotics-related research [1,2,3], demonstrating its effectiveness in various settings.
>
> [1] Sermanet, Pierre, et al. "Time-contrastive networks: Self-supervised learning from video." *2018 IEEE international conference on robotics and automation (ICRA)*. IEEE, 2018.
>
> [2] Li, Yunzhu, et al. "3d neural scene representations for visuomotor control." *Conference on Robot Learning*. PMLR, 2022.
>
> [3] Nair, Suraj, et al. "R3m: A universal visual representation for robot manipulation." *arXiv preprint arXiv:2203.12601* (2022).
>
> **Comment 2:**
>
> The 3D encoder-decoder model is trained on multi-view images, with scene representation ($z_t$ … ). How can it generalize when the inputs are from the same viewpoint, as in  ($z_t$ …) (line 291)?
>
> **Response 2:**
>
> The ability of the proposed model to perform single-view inference is enabled by the following factors (also described in the revised manuscript at lines 285-288).
>
> 1. **Time Contrastive Loss**: The time contrastive loss ensures that the state feature $v_t$ remains consistent regardless of which viewpoint image is provided for a given timestep $t$. This objective encourages the model to produce similar state features for all images corresponding to the same underlying scene state, even if the viewpoints differ.
> 2. **3D Geometry Awareness**: A fully 3D-aware encoder will output the same representation for any image of a scene at a specific timestep $t$, regardless of the viewpoint. To achieve this level of 3D geometry awareness, we employ training objectives such as cross-view completion and multi-view reconstruction. These objectives help the encoder learn the underlying 3D structure of the scene, ensuring consistent outputs for any combination of primary or reference inputs.
>
> While these individual factors do not achieve the desired property perfectly, they work complementarily, as demonstrated in the ablation study (Section 5.4).
>
> **Comment 3:**
>
> Table 1 claims that the proposed method does not require camera calibration. However, camera poses are needed to render multi-review reconstruction from $z_t$, making this claim inaccurate.
>
> **Response 3:**
>
> Thank you for your comment. First, we would like to kindly ask the reviewer to refer to the Common Response that addresses some misunderstandings regarding our work.
>
> Our work consists of 2 steps. 1) pre-train the 3D scene encoder to extract effective representation by leveraging NeRF (require calibrated camera at this phase for volume rendering), 2) perform inference on the pre-trained encoder and utilize the output representation of the encoder as an input for downstream RL. At this phase, we do **NOT** perform rendering (In Figure 1, the deployment phase does not include NeRF). Therefore, camera calibration is no longer required during the downstream RL phase.
>
> Since there is a similar comment (Comment 2 from reviewer JDLs),  we have modified Table 1 to separate the pre-training and deployment phases for a clearer comparison.
>
> Anonymous link for the modified table:
>
> https://drive.google.com/file/d/1fhHpJtBrUcDoLxycD13AnSD-aVv6RKX6/view?usp=sharing
>
>
>
> **Comment 4:**
>
> In the volume rendering experiments, the authors should also include comparisons with NeRF baselines for sparse views, such as RegNeRF, pixelNeRF, etc.
>
> **Response 4:**
>
> Thank you for your comments. As mentioned in the Common Response, we would like to clarify that the main goal of this work is to enhance downstream RL performance through NeRF-based representation learning, rather than achieving high-quality rendering or novel-view synthesis. As such, we believe that visual comparisons with NeRF variants like RegNeRF or pixelNeRF, which are optimized for sparse-view novel-view synthesis, are not directly aligned with the main objectives of this study.
>
> Even though we could consider integrating other NeRF models, we just utilized vanilla NeRF to focus on algorithmic-level development for representation learning. However, leveraging NeRF variants capable of better novel-view synthesis might improve the downstream RL performance under larger levels of viewpoint perturbation, which could be an interesting direction for future exploration.

---

> > ### Author Response · Authors · 2024-11-19
> > **Author Response 2**
> >
> > **Comment 5:**
> >
> > The RL experiments are conducted on toy environments. It would be valuable to see the method's performance in real-world robotic settings.
> >
> > **Response 5:**
> >
> > Real-world experiments are certainly feasible. However, as a 3D representation learning framework for RL has emerged only recently, there remain many challenges and open questions. Consequently, most of the prior 3D representation-based RL works (and our method) are more focused on developing algorithms for the effective 3D representation learning framework rather than deploying them directly in real-world settings. Therefore, we leverage simulations where the proposed algorithm can be extensively evaluated and analyzed in multiple environments.
> >
> > Our method, which supports single-view inference for downstream RL, is well-suited for future extensions into real-world applications. Specifically, we plan to address the requirement for synchronized multiple cameras during pre-training by exploring the use of videos captured from moving cameras with varying viewpoints (as mentioned in Section 6). This direction will enable seamless transitions to experiments with real robots, spanning pre-training to downstream online RL, while building on the strengths of the current proposed single-view inference framework.
> >
> > If you have any questions or need more discussion, please let us know. We would be happy to improve our work based on your valuable feedback.

---

> > > ### Author Response · Authors · 2024-11-24
> > > **Remind**
> > >
> > > We sincerely thank all the reviewers for reviewing our work and providing constructive feedback. We hope that our response has adequately addressed your comments. If you have any remaining questions (existing or new ones) that we can address in our follow-up response to improve your opinion about our work, please do not hesitate to provide additional feedback in the comments. It would be greatly appreciated if we could have more discussions about our work which would provide valuable insights towards further developing our research into a meaningful contribution in the RL domain.

---

> ### Comment · Reviewer_amkh · 2024-11-28
>
> Thank you to the authors for their response. They have adequately addressed my concerns, and I have no further questions.

---

> > ### Author Response · Authors · 2024-11-29
> >
> > We sincerely appreciate all your efforts during the review process. We believe we have addressed all the comments and hope the updated manuscript better reflects the quality and impact of our work. If our response is not sufficiently clear to reconsider the score, please do not hesitate to let us know. We would be happy to discuss further to improve our work.

---

### Official Review · Reviewer_4UuN · 2024-11-10

**Soundness:** 3
**Presentation:** 3
**Contribution:** 3
**Rating:** 3
**Confidence:** 4

**Summary:**

The paper proposes an interesting approach to utilize NeRF based pretraining to bake in viewpoint awareness into an RL system. The authors first pretrain a representation using cross-view completion objective visa NeRF rendering using time contrastive learning objective for scene regularization. The authors then use the pretrained scene encoder for downsteam reinforcement learning task. Relevant experiments are designed which demonstrate viewpoint awareness of the system in a synthetic setup.

**Strengths:**

In my opinion, below are the strengths of the approach:

1. Designing relevant experiments and showcasing improvement numbers that highlight the method is invariant to the viewpoint and camera matrices. Slight perturbation in the cameras from the reference views shows the learned policies are invariant to disturbances.

2. The writing and flow of the paper is nice, and the presentation is clear.

3. Strong qualitative improvement results against competing baselines.

**Weaknesses:**

In my opinion, the weakness of the paper is as follows:

1. The paper misses various key recent results both for 3D representation learning using NeRFs [1] and for baking in viewpoint awareness for policy learning [2,3]. In my opinion, the paper is incomplete without discussion or comparison to these approaches.

2. The paper doesn't show any real-world evaluation results while both [2,3] show real-world results. Is it an inherent limitation of the method that it only works in simulation?

3. Follow-up to point 1. While the paper shows qualitative comparison to recent NeRF-based methods, how does the result compare to zero-shot generalizable NeRF-based method i.e. ZeroNVS (zero-shot vs. finetuned on their data) and NeRF representation learning method i.e. NeRF-MAE trained on their data?

4. What is the pretraining data mix and how does it impact OOD policy learning? Can the model generalize to OOD in sim i.e. sim2sim generalization or OOD real i.e. sim2real generalization?

[1] Irshad et al., ECCV 2024 NeRF-MAE: Masked AutoEncoders for Self-Supervised 3D Representation Learning for Neural Radiance Fields
[2] Chen et al. CORL 2024, RoVi-Aug: Robot and Viewpoint Augmentation for Cross-Embodiment Robot Learning
[3] Tian et al. CORL 2024, View-Invariant Policy Learning via Zero-Shot Novel View Synthesis
[4] Sargent et al. ZeroNVS: Zero-Shot 360-Degree View Synthesis from a Single Real Image

**Questions:**

Please see my questions in the weakness section above. I look forward to author's responses.

---

> ### Author Response · Authors · 2024-11-19
> **Author Response 1**
>
> **Comment 1:**
>
> The paper misses various key recent results both for 3D representation learning using NeRFs [1] and for baking in viewpoint awareness for policy learning [2,3].
>
> **Response 1:**
>
> As you rightly commented, NeRF-MAE [1] learns 3D representation. However, this method is not suitable for the RL setup because the encoder’s input consists of voxels, **which must be obtained through a pre-trained NeRF model**. Specifically, 1) we must have an individual pre-trained NeRF model for each scene (corresponding to each timestep in our work), **even during the downstream task,** to compute the voxel’s 4-channel values. Additionally, 2) these 4-channel values are computed by averaging the outputs of the pre-trained NeRF model across all viewing directions, **which requires access to multi-view images** during inference. This reliance makes NeRF-MAE incompatible with the single-view inference requirements of the RL setup. Therefore, we believe that NeRF-MAE is not directly comparable to our method.
>
> Both RoVi-Aug [2] and VISTA [3] use ZeroNVS [4] for data augmentation by synthesizing novel-view images, focusing on generating novel-view data rather than learning effective representations. In contrast, our approach centers on pre-training an image encoder to extract effective 3D-aware representations for downstream RL tasks by leveraging NeRF, based on the assumption of a given multi-view dataset. Consequently, prior works [2, 3, 4] are not used as direct baselines. Rather, these methods can serve as **complementary** components, as we could replace the multi-view dataset assumption with a dataset augmented by ZeroNVS, as in [2, 3], to further enhance our approach.
>
> To provide preliminary insights, we conducted data augmentation experiments to evaluate the policy’s robustness to viewpoint changes, similar to [2,3]. As mentioned in Response 3, ZeroNVS does not produce reasonable zero-shot synthesis results. Therefore, we used ground truth images obtained from simulation as a proxy for the data augmentation effect of ZeroNVS (i.e., assuming ZeroNVS synthesizes 100% accurate images). Specifically, we performed an RL experiment similar to Section 5.3, while following CNN+view randomization described in the manuscript, but with 30 viewpoints. The results are available in the following anonymous link:
>
> https://drive.google.com/drive/folders/1oUNOOcUNQGkIcCjlpFKT_B5tyhr-HJ3J?usp=sharing
>
> As shown in these figures, our method outperforms this baseline despite utilizing images from significantly fewer viewpoints (Ours: 6 views, Baseline: 30 views). We attribute this result to the proposed 3D-aware representation learning scheme, which enhances the encoder’s implicit understanding of the 3D world. This finding underscores that simply increasing the number of viewpoints is not always the optimal approach; instead, carefully designed representation learning schemes play a far more critical role.
>
> **Comment 2:**
>
> The paper doesn't show any real-world evaluation results while both [2,3] show real-world results. Is it an inherent limitation of the method that it only works in simulation?
>
> **Response 2:**
>
> This is not an inherent limitation, and real-world experiments are certainly feasible. However, as a 3D representation learning framework for RL has emerged only recently, there remain many challenges and open questions. Consequently, most of the prior 3D representation-based RL works (and our method) are more focused on developing algorithms for the effective 3D representation learning framework rather than deploying them directly in real-world settings. Therefore, we leverage simulations where the proposed algorithm can be extensively evaluated and analyzed in multiple environments.
>
> Our method, which supports single-view inference for downstream RL, is well-suited for future extensions into real-world applications. Specifically, we plan to address the requirement for synchronized multiple cameras during pre-training by exploring the use of videos captured from moving cameras with varying viewpoints (as mentioned in Section 6). This direction will enable seamless transitions to experiments with real robots, spanning pre-training to downstream online RL, while building on the strengths of the current proposed single-view inference framework.

---

> > ### Author Response · Authors · 2024-11-19
> > **Author Response 2**
> >
> > **Comment 3:**
> >
> > Follow-up to point 1. While the paper shows qualitative comparison to recent NeRF-based methods, how does the result compare to zero-shot generalizable NeRF-based method i.e. ZeroNVS (zero-shot vs. finetuned on their data) and NeRF representation learning method i.e. NeRF-MAE trained on their data?
> >
> > **Response 3:**
> >
> > First, we would like to kindly ask the reviewer to refer to the Common Response that addresses some misunderstandings regarding our work. The purpose of comparing rendering results with other 3D RL baselines in our work (such as SNeRL, NeRF-RL, etc) was to indirectly verify whether the learned 3D representation effectively captures 3D scene information. Achieving high-quality rendering in novel-view was not the primary objective of our work and the baselines.
> >
> > Following the reviewer’s comment, we visualize the zero-shot results of ZeroNVS (without fine-tuning) by distilling a NeRF for each environment at a specific episode and timestep.
> >
> > Anonymous link for ZeroNVS (without fine-tuning) results:
> >
> > [https://drive.google.com/file/d/1fhHpJtBrUcDoLxycD13AnSD-aVv6RKX6/view?usp=sharing](https://drive.google.com/file/d/1hJrYzAyH8YwFYggKd7vB1yB7IrQE0K9H/view?usp=drive_link)
> >
> > Even though ZeroNVS is capable of synthesizing moderately 3D consistent images near the input camera viewpoint, rendering quality significantly degrades and 3D scene structures are distorted as the azimuth angle varies more than 15 degrees. This suggests that using ZeroNVS as a data augmentation strategy to follow the reviewer’s comment would require access to more than a single viewpoint to generate a sufficient number of augmented images with acceptable quality across diverse viewpoints.
> >
> > Even if ZeroNVS with fine-tuning could improve the rendering quality, (1) policy learning with this novel-view synthesis-based data augmentation would be a complementary approach, as mentioned in Response 1, and (2) performing NeRF distillation process at every episode and timestep would be computationally expensive and extremely time-consuming, as noted as a limitation in VISTA.
> >
> > Regarding NeRF-MAE, regardless of its rendering quality, this method is not suitable for the RL setup, as mentioned in Response 1. Even if we just want to test the visualization, the pre-trained model checkpoints are not available and training from scratch would require significant computational resources and time to train a NeRF model for each episode and timestep. On the contrary, our method does not require individual NeRF models for each different timestep and episode.
> >
> > **Comment 4:**
> >
> > What is the pretraining data mix and how does it impact OOD policy learning? Can the model generalize to OOD in sim i.e. sim2sim generalization or OOD real i.e. sim2real generalization?
> >
> > **Response 4:**
> >
> > Based on the prior comments, we believe that the reviewer is inquiring about generalization in the context of the pre-training setup similar to NeRF-MAE, which uses 3D scene datasets containing diverse, mixed environments. However, we would like to clarify that our approach trains a separate 3D scene encoder for each environment, following the prior works on representation learning in RL. We clarify this in the revised manuscript (line 307-308). Furthermore, the lack of a large-scale 3D dataset specifically tailored for robotics poses a significant limitation for pursuing such generalization-focused research at this stage.
> >
> > To provide preliminary insights, we conducted experiments where the RL policy trained with our method was tested under conditions involving color variations to diversify visual appearances.
> >
> > Anonymous link to the color variation experiments:
> >
> > https://drive.google.com/drive/folders/1bjKhgwDSCFcnyLWTYSrq5x1nKYR7deIW?usp=sharing
> >
> > The results demonstrated slight generalization capabilities, even without explicit consideration of these variations. This might be due to the training strategy that encourages 3D geometry-awareness.
> >
> > If you have any questions or need more discussion, please let us know. We would be happy to improve our work based on your valuable feedback.

---

> > > ### Author Response · Authors · 2024-11-24
> > > **Remind**
> > >
> > > We sincerely thank all the reviewers for reviewing our work and providing constructive feedback. We hope that our response has adequately addressed your comments. If you have any remaining questions (existing or new ones) that we can address in our follow-up response to improve your opinion about our work, please do not hesitate to provide additional feedback in the comments. It would be greatly appreciated if we could have more discussions about our work which would provide valuable insights towards further developing our research into a meaningful contribution in the RL domain.

---

> > > > ### Comment · Reviewer_4UuN · 2024-12-02
> > > > **Thanks for your response**
> > > >
> > > > Thanks to the author for a detailed response to my questions and providing additional insights. I don't have any additional questions.

---

### Author Response · Authors · 2024-11-19
**Common Response**

We sincerely thank all the reviewers for reviewing our work and providing constructive feedback. We would like to clarify a few common misunderstandings regarding our work.

1. This work is **NOT** about how to obtain high-quality single-view 3D reconstruction or novel view synthesis. Instead, we focus on how to obtain a good representation for image-based RL, which itself is a huge field in RL. Our answer is to encourage 3D understanding of the encoder $\Omega_\theta$. And, the contribution is obtaining such representation only with single-view RGB input, while other prior representation learning works in RL require multi-view inputs with camera pose (Table 1).
2. The camera calibration is required during the 3D scene encoder pre-training since it requires volume rendering via NeRF. However, during the downstream RL, we only perform inference on the pre-trained encoder and do **NOT** perform rendering (In Figure 1, the deployment phase does not include NeRF). Therefore, camera calibration information is no longer required during the downstream RL phase.

We acknowledge that the current manuscript seems slightly focused on the single-view reconstruction results, which we think are indirect validations for our learned latent representation $z_t$’s 3D understanding. We have uploaded a revised manuscript with the magenta color highlight, which has the following key changes.

1. We have modified the introduction, and experiment section to make clear that our work has contributions in representation learning for RL, not in a high-quality novel-view synthesis.
2. We have clarified that our work consists of 2 stages (pre-training, deployment) and does not require NeRF rendering (and calibrated camera) during the downstream RL deployment.
3. We have replaced the term ‘3D representation’ with ‘3D-aware representation’ since someone might misunderstand that it corresponds to some explicit 3D representation in the computer vision field.

We further described the details for the above points in responses for each reviewer, so please kindly refer to our responses.

---

> ### Author Response · Authors · 2024-11-29
> **Minor**
>
> We have updated some of the attached anonymous links as there were access issues with the previous ones. Please let us know if you still encounter any problems accessing the links.

---

### Meta-Review · Area_Chair_8KLE · 2024-12-20

**Metareview:**

This work examines image-conditioned RL-based policy learning, and the focus is to obtain 3D-aware image representation that can be used as input for policy training. A combination of contrastive learning and cross-view prediction objectives ensure that the encoded feature is 3D-aware, and the experiments across different tasks show that these features allow more efficient learning with higher performance.

The reviewers appreciated the empirical improvements and the robustness of the learned policies. However, there were some concerns about prior 3D-aware RL work (though the author response highlighted the single-view nature of deployment compared to prior work).  There were also concerns about the practical applications and generalization as the experiments are all in synthetic settings, with each model trained and tested in the same environment.

On the balance, these issues outweighed the benefits, and the reviewers leaned towards rejection. The AC also agrees with this sentiment, and in particular is swayed by the limitations in practical benefits and limited generalizability. Specifically, given that the training and testing is in the same environment, requiring multi-view input at training, it is not clear why this cannot be practically assumed at inference  — this would not have been a concern had the paper trained/tested across different environments (but then the single latent variable would perhaps not have sufficed for a conditional NeRF input). The authors are encouraged to expand this work in terms of generalization and potential real-world applications to truly highlight the benefits of the framework.

**Additional Comments On Reviewer Discussion:**

The reviewers raised concerns about prior 3D-aware RL work, the quality of the 2D renderings, real-world applications, and limited generalization ability. The author response addressed some of these by pointing out that this approach requires single-view input at deployment (unlike prior multi-view methods) and that the rendering quality was not the prime focus. While the author response also addressed several of the questions raised, it ultimately did not sway the opinions of the reviewers to be more positive. The AC also agrees with some of these concerns raised, in particular the practical benefits and the limited generalization.

---

### Decision · Program_Chairs · 2025-01-22

Reject